# Methionine restriction breaks obligatory coupling of cell proliferation and death by an oncogene Src in *Drosophila*

**Hiroshi Nishida[1,2], Morihiro Okada[2,3†], Lynna Yang[2†], Tomomi Takano[2,3], Sho Tabata[4], Tomoyoshi Soga[4], Diana M Ho[5], Jongkyeong Chung[6], Yasuhiro Minami[1], Sa Kan Yoo[2,3,7]\***

[1]Division of Cell Physiology, Kobe University, Kobe, Japan; [2]RIKEN CPR, Kobe, Japan; [3]RIKEN BDR, Kobe, Japan; [4]Institute for Advanced Biosciences, Keio University, Tsuruoka, Japan; [5]Harvard Medical School, Boston, United States; [6]Seoul National University, Seoul, Republic of Korea; [7]Division of Developmental Biology and Regenerative Medicine, Kobe University, Kobe, Japan

**Abstract** Oncogenes often promote cell death as well as proliferation. How oncogenes drive these diametrically opposed phenomena remains to be solved. A key question is whether cell death occurs as a response to aberrant proliferation signals or through a proliferation-independent mechanism. Here, we reveal that Src, the first identified oncogene, simultaneously drives cell proliferation and death in an obligatorily coupled manner through parallel MAPK pathways. The two MAPK pathways diverge from a lynchpin protein Slpr. A MAPK p38 drives proliferation whereas another MAPK JNK drives apoptosis independently of proliferation signals. Src-p38-induced proliferation is regulated by methionine-mediated Tor signaling. Reduction of dietary methionine uncouples the obligatory coupling of cell proliferation and death, suppressing tumorigenesis and tumor-induced lethality. Our findings provide an insight into how cells evolved to have a fail-safe mechanism that thwarts tumorigenesis by the oncogene Src. We also exemplify a diet-based approach to circumvent oncogenesis by exploiting the fail-safe mechanism.

**\*For correspondence:**
sakan.yoo@riken.jp

[†]These authors contributed equally to this work

**Competing interests:** The authors declare that no competing interests exist.

## Introduction

Tumorigenesis requires activation and inactivation of not one, but multiple signaling pathways (*Hanahan and Weinberg, 2011*; *Lowe et al., 2004*; *Shortt and Johnstone, 2012*). This is likely because cells have evolutionarily refined fail-safe mechanisms to prevent tumorigenesis by a single oncogene activation. Because of this fail-safe mechanism, the transforming effects of oncogenes are often cancelled by the cell's intrinsic ability to prevent tumorigenesis (*Lowe et al., 2004*; *Shortt and Johnstone, 2012*). Thus, transformation by oncogenic drivers requires inhibition of tumor suppressor signaling, which regulates cell death or senescence.

A well-known example of the fail-safe mechanism is the case of *Myc* oncogene. *Myc* is a potent oncogene implicated in most human tumor entities, but it is also a powerful cell death driver (*Evan et al., 1992*; *Hoffman and Liebermann, 1998*; *Prendergast, 1999*). Originally, two models were proposed to explain the Myc-induced cell death (*Evan et al., 1994*). The 'conflict' model suggests that, in an environment where proliferation is not supported, inappropriate growth signals induced by Myc invoke the cell's intrinsic mechanisms to cease abnormal proliferation, leading to apoptosis. In this model, the main function of Myc is to induce proliferation, and cells undergo apoptosis in response to the aberrant growth signal. In contrast, the 'dual' model proposes that apoptosis occurs as a direct result of bona fide signaling of the Myc pathway, rather than as the cell's response to Myc-invoked aberrant growth signaling. In this model, cell death is a normal, obligate

function of Myc, which is intrinsically imprinted in the Myc signaling itself. Although these two models are not mutually exclusive, delineation of Myc signaling supports the dual model: Myc drives p53-mediated apoptosis through transcriptional regulation of the tumor suppressor ARF (*Lowe et al., 2004*; *Shortt and Johnstone, 2012*).

In addition to Myc, many other oncogenes are known to induce cell death. This phenomenon is now accepted as an intrinsic tumor suppressive mechanism (*Shortt and Johnstone, 2012*). However, contrary to the well-studied mechanisms of Myc-induced cell death, how other oncogenes, such as *Ras* or *Src*, couple cell proliferation and death remains unclear (*Lowe et al., 2004*).

In this study, we focus on the *Src* oncogene. *Src* is the first oncogene identified (*Yeatman, 2004*). Src expression and activity is often increased in human cancer, which contributes to oncogenesis (*Ishizawar and Parsons, 2004*; *Summy and Gallick, 2003*; *Yeatman, 2004*). Due to the implication of Src in tumorigenesis, many clinical inhibitors targeting Src family kinases (SFKs) have been developed but their use as therapeutic drugs has been unsuccessful (*Gargalionis et al., 2014*; *Sousa-Victor and Jasper, 2014*). The reason for the low efficacy of these Src inhibitors remains unclear because both SFKs and SFK inhibitors have a broad spectrum of targets, making mechanistic analyses difficult. This compels us to better understand how Src regulates signaling pathways.

*Drosophila* has two SFKs: *Src42A* and *Src64B* (*Kussick et al., 1993*; *Pedraza et al., 2004*; *Takahashi et al., 1996*). The endogenous expression patterns of *Src42A* and *Src64*B are different, but ectopic expression of either of them induces similar effects. *Drosophila* Src regulates a variety of signaling pathways, including Notch, MAPKs, Jak-Stat, EGF, Wnt, and Hippo signaling (*Cordero et al., 2014*; *Enomoto and Igaki, 2013*; *Ho et al., 2015*; *Read et al., 2004*; *Tateno et al., 2000*). Src activation induces apoptosis as well as tissue growth (*Pedraza et al., 2004*). However, how cell death and proliferation are coordinated downstream of Src activation still remains elusive. Here, we investigate how the *Src* oncogene couples cell proliferation and death in epithelia of the wing imaginal disc.

## Results

Previous studies have shown that a *Drosophila* Src, Src42A, simultaneously induces cell death and proliferation in both eye and wing imaginal discs (*Pedraza et al., 2004*). In this study, we used a *vg-Gal4* driver, which promotes local expression in the DV boundary and a part of the hinge region of the wing disc. Indeed, expression of constitutively active (CA) *Src42A* with an amino acid substitution of Tyr$^{511}$ to Phe, which is refractory to inactivating phosphorylation by Csk (*Tateno et al., 2000*), induced both mitosis and apoptosis, which were detected by phospho-histone 3 (pH3) and cleaved caspase DCP1, respectively (*Figure 1A–D*). This provocation of both cell proliferation and death results in a mild overgrowth in the *Src42A CA*-expressing region, likely reflecting the mutually cancelling effects of proliferation and death. (*Figure 1E*).

Src signaling mediates a myriad of pathways, but how these signals converge to produce specific phenotypes remains unclear, especially in the context of cell proliferation and death coupling. It has previously been shown that Src-induced cell death cannot be inhibited by *p21* overexpression, which suppresses cell proliferation (*Pedraza et al., 2004*). But, since p21 directly inhibits the cell cycle through inhibition of Cyclin-dependent kinases, the most downstream component of proliferation signals, it still remains unclear whether Src-induced aberrant proliferation signals play a role in cell death.

In order to find how Src signaling drives both cell proliferation and death simultaneously and whether cell death occurs in response to proliferation signals, we reasoned that interrogation of downstream signaling provoked by Src activation will give us clues on how these two opposing phenotypes are driven by Src. To search for an effector downstream of Src, we performed an RNAi screening mainly focusing on cell death-related factors (*Supplementary file 1*). The screening took advantage of the organismal lethality induced by *Src42A CA* expression in the wing disc (*Figure 2A*). We searched for RNAis that could suppress this Src-induced lethality.

We identified that *slipper* (*slpr*), a mixed lineage kinase, regulates Src signaling. Src-induced lethality was suppressed by *slpr* knockdown (*Figure 2A* and *Figure 2—figure supplement 1A–B*). We confirmed knockdown efficiency of the *slpr* RNAi (*Figure 2—figure supplement 1C*) by RT-qPCR. Rare escapers of *vg>Src42A CA* flies have small, disheveled wings, but *slpr* knockdown completely reversed this phenotype (*Figure 2B*, *Figure 2—figure supplement 1A*). This is not an

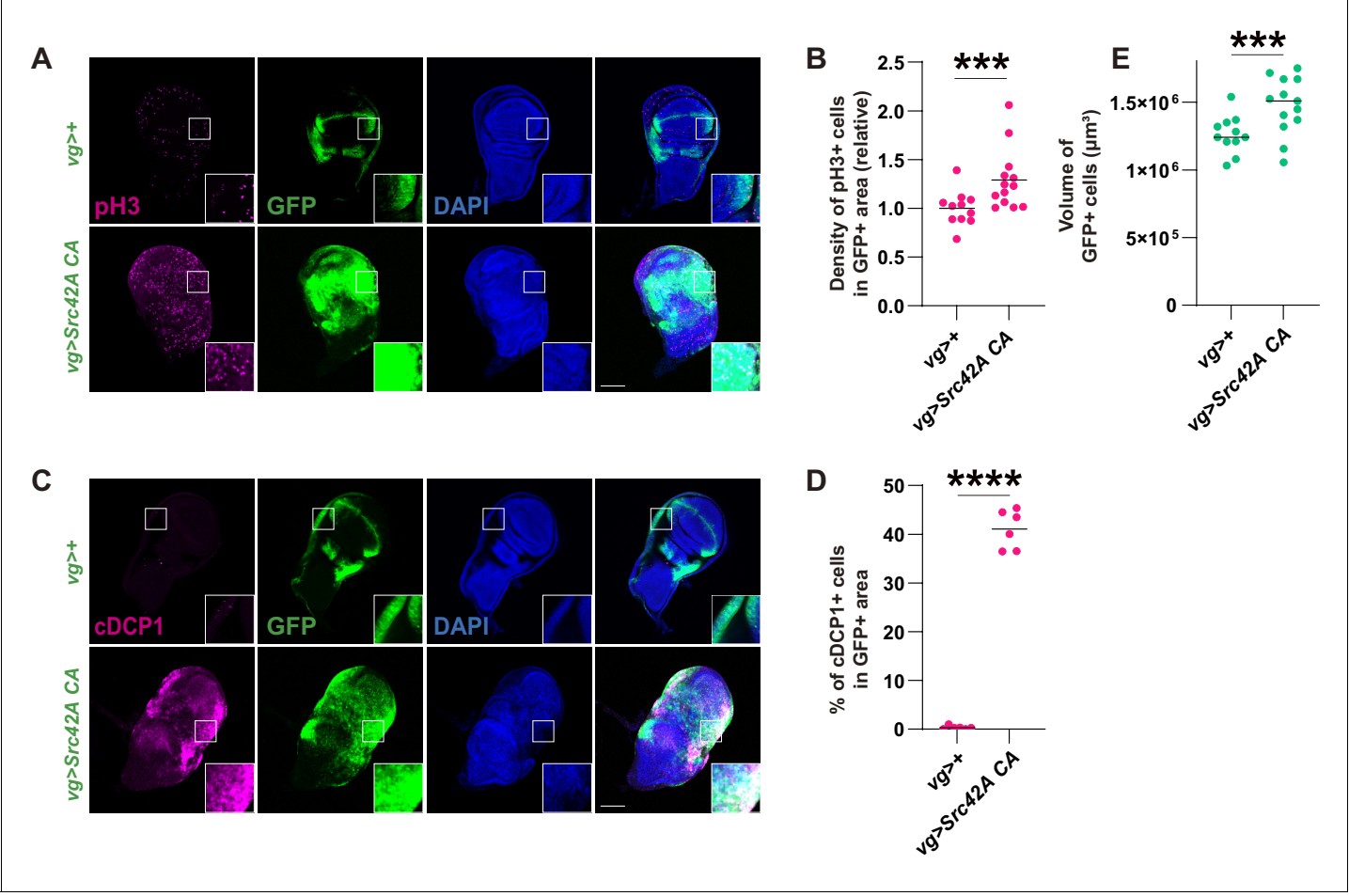

**Figure 1.** Src activation induces both cell proliferation and death, resulting in a mild tissue overgrowth. (A) *Src42A constitutively active* (*CA*) expression induces cell proliferation, which was detected by phospho-histone 3 (pH3) staining. (B) Quantification of pH3 staining. The number of pH3+ cells was normalized by the area of GFP+ cells. Two-tailed unpaired t-test. (C) *Src42A CA* expression induces caspase activation, which was detected by cleaved DCP1 staining. (D) Quantification of percentage of cDCP1+ cells in GFP+ cells. Two-tailed unpaired t-test. (E) Quantification of the total volume of GFP + cells (µm$^3$). Two-tailed unpaired t-test. Scale bars, 100 µm.

event that is observed only in the wing disc because Slpr inhibition in the eye disc also suppressed organismal lethality and the abnormal eye phenotype of escapers that are induced by the *Src* expression in the eye disc (*Figure 2—figure supplement 1D–E*).

The link between Src and Slpr has not been previously known. Slpr was originally identified as a JNKKK, a MAPKKK that regulates JNK signaling (*Stronach and Perrimon, 2002*). Src has long been known to regulate JNK signaling (*Tateno et al., 2000*). Some effectors such as ubiquitin E2 complex Bendless (*Ma et al., 2013*) and F-actin cytoskeleton (*Enomoto and Igaki, 2013*; *Fernández et al., 2014*; *Rudrapatna et al., 2014*) have been shown to link Src and JNK. But how Src exactly regulates JNK signaling still remains elusive. We speculated that Slpr may link Src and JNK signaling. Indeed, *slpr* RNAi significantly suppressed Src-mediated activation of JNK, which was detected by the JNK activity reporter TRE-RFP (*Figure 2C–D*). This observation suggests that Slpr is a JNKKK that transduces Src activation to JNK signaling.

Although our findings that Slpr mediates Src-induced JNK activation are consistent with the previous literature showing that Slpr is a MAPKKK upstream of JNK, the effects of *slpr* knockdown on the wings and organismal survival over Src stress were perplexing. This is because a combination of JNK inhibition and Src activation is known to induce a massive overgrowth phenotype (*Enomoto and Igaki, 2013*; *Ho et al., 2015*). Indeed, simultaneously activating Src and suppressing JNK resulted in substantial overgrowth of the wing disc (*Figure 3A–D*) and complete organismal lethality

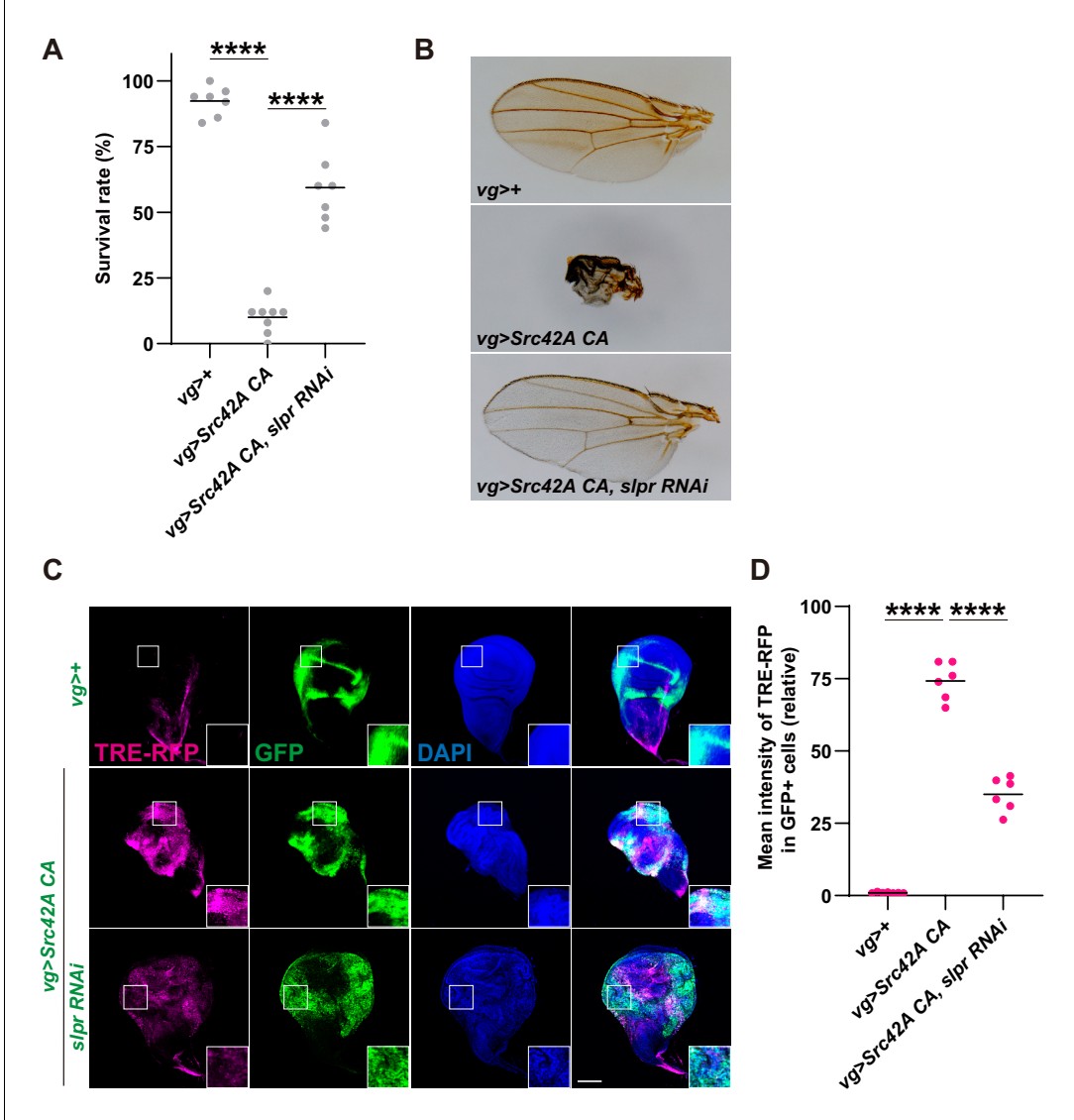

**Figure 2.** Inhibition of Slpr suppresses the phenotypes induced by Src activation. (**A**) *Src42A constitutively active (CA)* expression in the wing disc induces organismal lethality, which is suppressed by knockdown of *slpr*. One-way ANOVA with Sidak's post-test. (**B**) The small, disheveled wing phenotype of the rare escapers with Src42A CA is suppressed by knockdown of *slpr*. (**C**) Src42A CA-mediated JNK activation, which was detected by the TRE-RFP reporter, is suppressed by knockdown of *slpr*. (**D**) Quantification of TRE-RFP in C. One-way ANOVA with Sidak's post-test. Scale bars, 100 μm.

The online version of this article includes the following figure supplement(s) for figure 2:

**Figure supplement 1.** Inhibition of Slpr suppresses the phenotypes induced by Src activation.

(*Figure 3E*). On the other hand, combining Src activation and Slpr inhibition did not induce tissue overgrowth (*Figure 3A–D*). We found that Slpr inhibition suppressed both cell proliferation and apoptosis induced by Src whereas JNK inhibition suppressed only apoptosis but not cell proliferation (*Figure 3A–D*). We confirmed the same results with a different RNAi for *slpr* (*Figure 3—figure supplement 1A–D*). This suggests that JNK inhibition-mediated suppression of apoptosis is responsible for the tumor overgrowth. Consistent with this idea, a combination of Src activation and apoptosis inhibition by microRNAs for reaper, hid and grim (*Siegrist et al., 2010*), which inhibits DIAP1, also induced overgrowth, similar to JNK inhibition (*Figure 3—figure supplement 1E*). On the other hand, a combination of Src activation and cell death induction enhanced survival over the Src tumor (*Figure 3—figure supplement 1F*). Furthermore, the overgrowth phenotype induced by Src

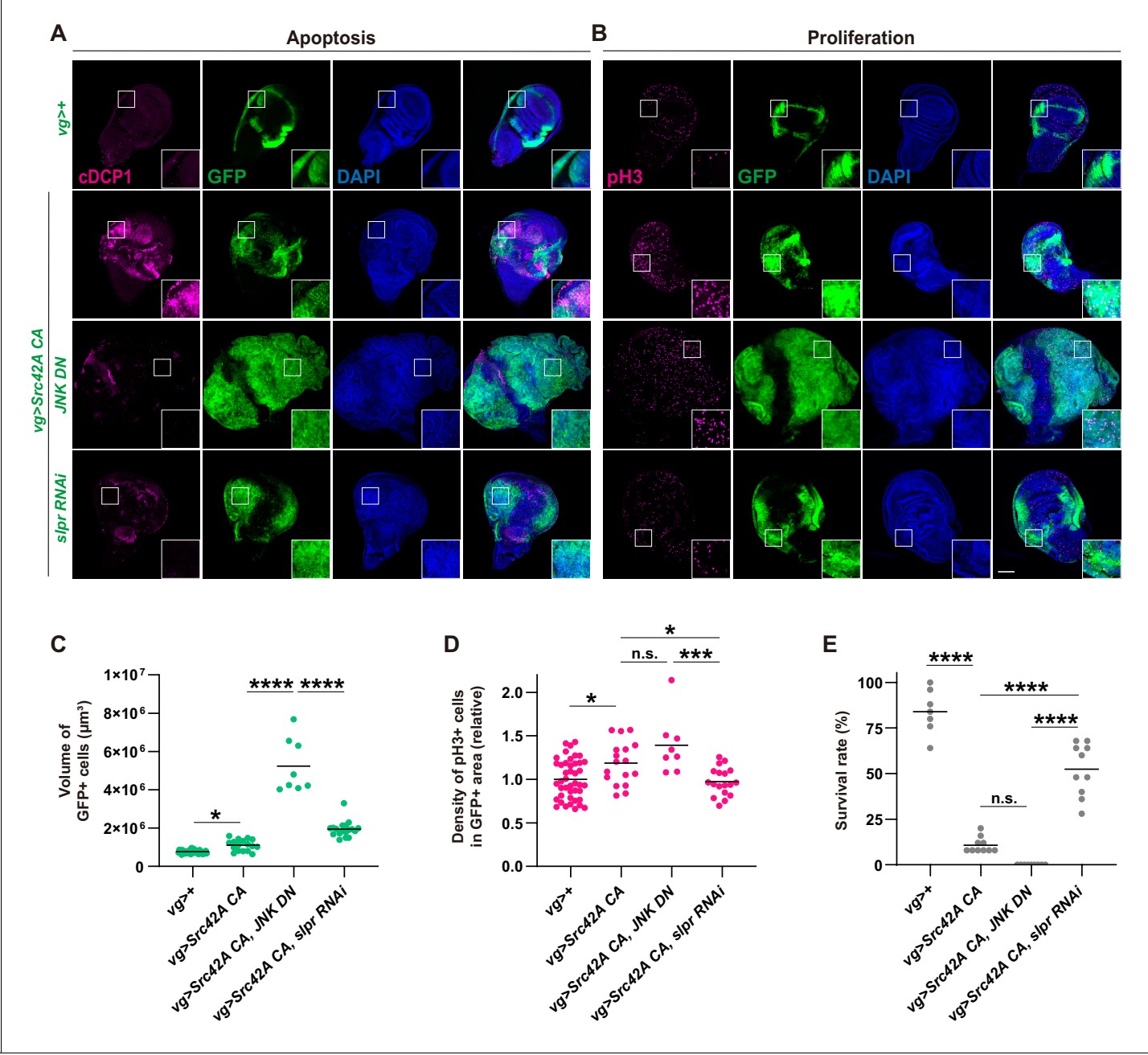

**Figure 3.** Slpr regulates both cell proliferation and cell death that are induced by Src activation. (A) Apoptosis induced by Src42A constitutively active (CA) is suppressed by *slpr* RNAi and JNK DN. Note the aggressive overgrowth phenotype induced by combining Src with JNK DN. (B) JNK DN does not inhibit Src42A CA-mediated proliferation whereas slpr inhibition does. (C) Quantification of the total volume of GFP+ cells ($\mu m^3$) in B. One-way ANOVA with Sidak's post-test. (D) Quantification of phospho-histone 3 (pH3) staining in B. The number of pH3+ cells was normalized by the area of GFP+ cells. Src42A CA-induced proliferation is suppressed by knockdown of *slpr* but not by overexpression of *JNK DN*. One-way ANOVA with Sidak's post-test. (E) Inhibition of JNK enhances organismal lethality induced by Src42A CA. One-way ANOVA with Sidak's post-test. Scale bars, 100 μm. The online version of this article includes the following figure supplement(s) for figure 3:

**Figure supplement 1.** Inhibition of Slpr suppresses the phenotypes induced by Src activation.

activation and JNK inhibition was suppressed by *slpr* knockdown (*Figure 3—figure supplement 1G–I*). Taken together, these findings strongly imply that Slpr, which was originally identified as a MAPKKK upstream of JNK, regulates other signaling pathways along with the JNK pathway.

We investigated what is regulating proliferation downstream of the Src-Slpr axis if JNK only regulates cell death. Although Slpr was originally identified as a JNK regulator (*Stronach and Perrimon, 2002*), subsequent studies report the existence of situations where Slpr can also activate other MAPKs including Erk and p38 (*Chen et al., 2010*; *Sathyanarayana et al., 2003*). Thus, we examined whether p38 and Erk could function as downstream components of Src-Slpr signaling. We found that Src activates both Erk and p38 in the wing disc (*Figure 4A–B* and *Figure 4—figure supplement 1A–C*). Activated Erk and p38 were observed not only in the region that expresses *Src*, but also in the surrounding area together with non-cell autonomous proliferation (*Figure 4A–B*, *Figure 4—*

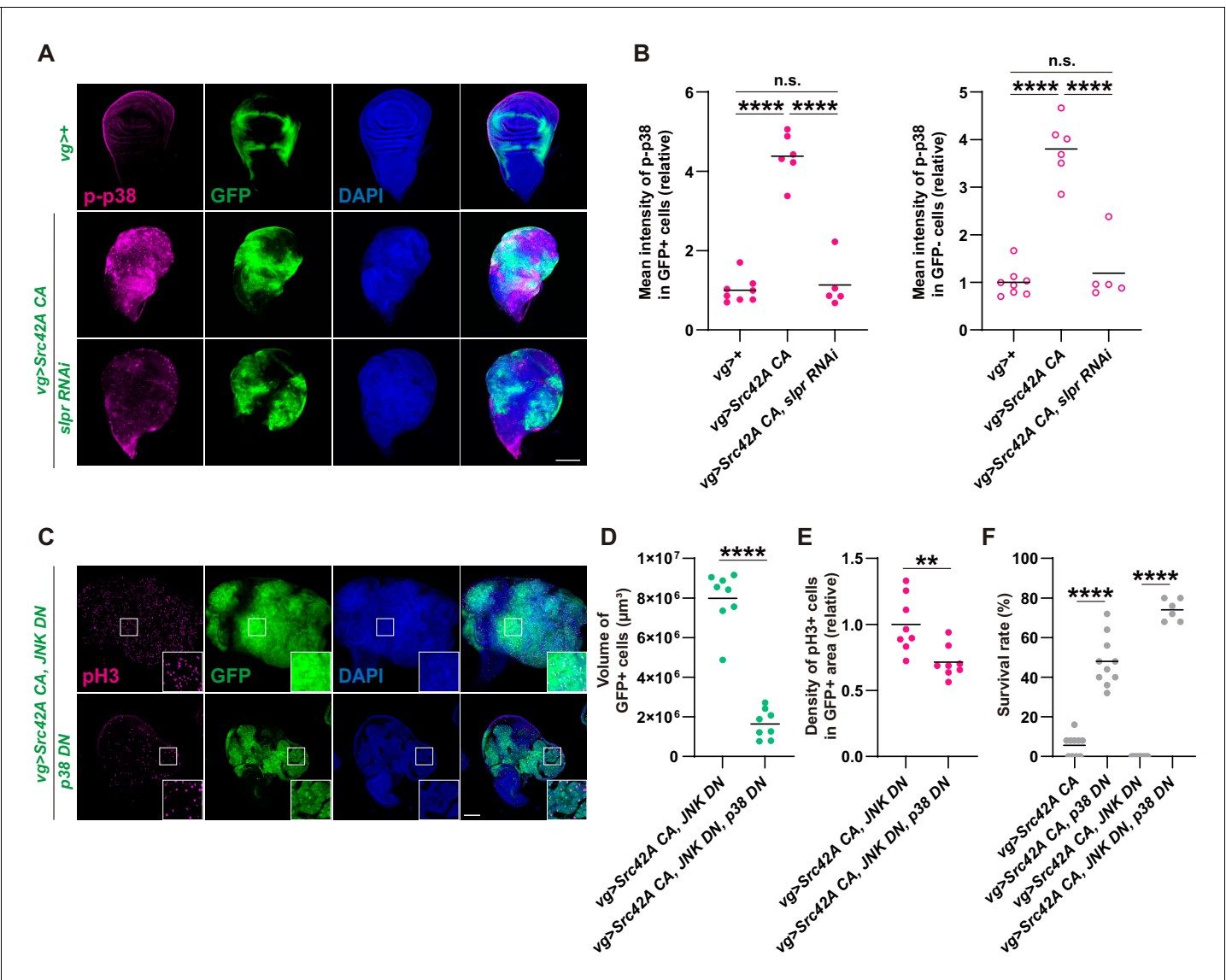

**Figure 4.** p38 mediates Src-induced cell proliferation. (A) *Src42A constitutively active* (*CA*) expression induces phosphorylation of p38 both cell autonomously and non-cell autonomously, which is suppressed by *slpr* knockdown. (B) Quantification of phosphorylated p38 in A. One-way ANOVA with Sidak's post-test. (C) Src42A CA-induced proliferation with/without JNK inhibition is suppressed by p38 DN. (D) Quantification of the total volume of GFP+ cells ($\mu m^3$) in C. Two-tailed unpaired t-test. (E) Quantification of phospho-histone 3 (pH3) staining in C. The number of pH3+ cells was normalized by the area of GFP+ cells. Two-tailed unpaired t-test. (F) Inhibition of p38 suppresses organismal lethality induced by Src42A CA. One-way ANOVA with Sidak's post-test. Scale bars, 100 μm.

The online version of this article includes the following figure supplement(s) for figure 4:

**Figure supplement 1.** Erk is activated downstream of Src-Slpr signaling but does not mediate cell proliferation.

**Figure supplement 2.** p38 inhibition does not inhibit Src-induced apoptosis.

*figure supplement 1A–D*). In fact, there was also non-cell autonomous activation of JNK and cell death, albeit much subtler than Erk and p38 (*Figure 4—figure supplement 1E–G*). This cell autonomous and non-cell autonomous activation of MAPKs by Src is similar to the activation patterns of Yorkie by Src (*Enomoto and Igaki, 2013*). Inhibition of Slpr reduced Src-induced activation of both p38 and Erk (*Figure 4A–B* and *Figure 4—figure supplement 1A–B*) cell autonomously, indicating that Slpr activates p38 and Erk in addition to JNK. It is of note that Slpr inhibition suppressed Src-induced non-cell autonomous activation of p38 but not Erk or JNK (*Figure 4B*, *Figure 4—figure supplement 1A–C,E*), suggesting that there are different mechanisms that regulate non-cell autonomous activation of these MAPKs. In line with persistence of the non-cell autonomous JNK activation with Slpr inhibition, Slpr inhibition did not affect non-cell autonomous activation of caspase (*Figure 4—figure supplement 1E, F, G*). This suggests that Src-invoked proliferation does not occur as a compensatory proliferation induced by death of the surrounding cells.

Next, we investigated which of p38 or Erk might be involved in Src-induced cell proliferation. When we inhibited Erk using several RNAis that had previously been utilized (*Singh et al., 2016*), Erk inhibition was unable to reverse Src-induced organismal lethality and proliferation (*Figure 4—figure supplement 1H–I*). We also verified that the *erk* RNAi we used could inhibit the rough eye phenotype induced by Ras activation (*Figure 4—figure supplement 1J*). This indicates that, even though Src-Slpr signaling activates Erk, its functional contribution to proliferation is negligible. In fact, Erk inhibition lowered survival over *Src* expression, implying that Src-Slpr-Erk signaling is protective for animals through an unknown mechanism. On the other hand, when we inhibited p38 in the *Src*-expressing region, it suppressed cell proliferation (*Figure 4C–E*), regardless of the existence of JNK inhibition (*Figure 4—figure supplement 2A–C*). Furthermore, inhibition of the p38 MAPKK, Licorne (Lic) (*Cully et al., 2010*), suppressed Src-induced proliferation (*Figure 4—figure supplement 2D–F*). p38 inhibition also suppressed Src-induced lethality (*Figure 4F*). Importantly, p38 suppression did not inhibit Src-mediated apoptosis (*Figure 4—figure supplement 2G–I*). We conclude that, downstream of Src-Slpr, p38 regulates cell proliferation without affecting apoptosis.

Since Src activates both proliferation and cell death through two MAPK signaling pathways emanating from Slpr, we reasoned that inhibition of p38-mediated proliferation will be beneficial for a potential therapeutic purpose, uncoupling the link between cell death and proliferation. However, since p38 itself could still activate multiple signaling pathways, we decided to further delineate how p38 regulates cell proliferation, aiming to find a way to uncouple proliferation and cell death in a specific manner.

First we focused on Mef2 and Atf2, the transcription factors that are known to be regulated by p38 (*Adachi-Yamada et al., 1999*; *Sano et al., 2005*; *Vrailas-Mortimer et al., 2011*). Using the previously published RNAis for *mef2* or *atf2* (*Clark et al., 2013*; *Terriente-Félix et al., 2017*) did not suppress aggressive tumorigenesis induced by simultaneous Src activation and JNK inhibition (*Figure 5—figure supplement 1A*).

We next shifted our focus to Tor signaling since p38 also regulates cell proliferation through Tor (*Cully et al., 2010*). Tor is a central growth regulator that transduces nutritional information to cell growth (*Kim and Guan, 2019*; *Sabatini, 2017*). We examined whether Src activates Tor signaling. Phosphorylation of 4EBP, a readout of Tor signaling activation, was enhanced by *Src* expression (*Figure 5A–B*). Src also promoted phosphorylation of S6, another readout of Tor signaling (*Kim et al., 2017*, *Figure 5—figure supplement 1B–C*). Src-induced phosphorylation of 4EBP and of S6 could be suppressed by Slpr or p38 inhibition, but not by JNK inhibition (*Figure 5A–B*, *Figure 5—figure supplement 1B–C*), indicating that Tor signaling functions downstream of the Src-Slpr-p38 axis. We then investigated whether Tor inhibition can suppress the massive overgrowth induced by Src activation and JNK inhibition. Tor inhibition completely suppressed overgrowth induced by Src activation and JNK inhibition (*Figure 5C*). Taken together, these findings indicate that p38 regulates cell proliferation through Tor signaling.

Tor functions as a nutrient sensor, which can be regulated by manipulation of diet organisms eat. Since dietary manipulation of nutrients, including sugar and amino acids, affects clinical cancer outcome (*Badgley et al., 2020*; *DeBerardinis and Chandel, 2016*; *Gao et al., 2019*; *Goncalves et al., 2019*; *Gonzalez et al., 2018*; *Kanarek et al., 2018*; *Knott et al., 2018*; *Maddocks et al., 2013*; *Maddocks et al., 2017*; *Pavlova and Thompson, 2016*), we investigated whether dietary manipulation of nutrition can mimic the phenotype of Tor or p38 inhibition. We found that simple dilution of yeast in the fly food suppressed the tumor overgrowth and Tor activation that are induced by Src

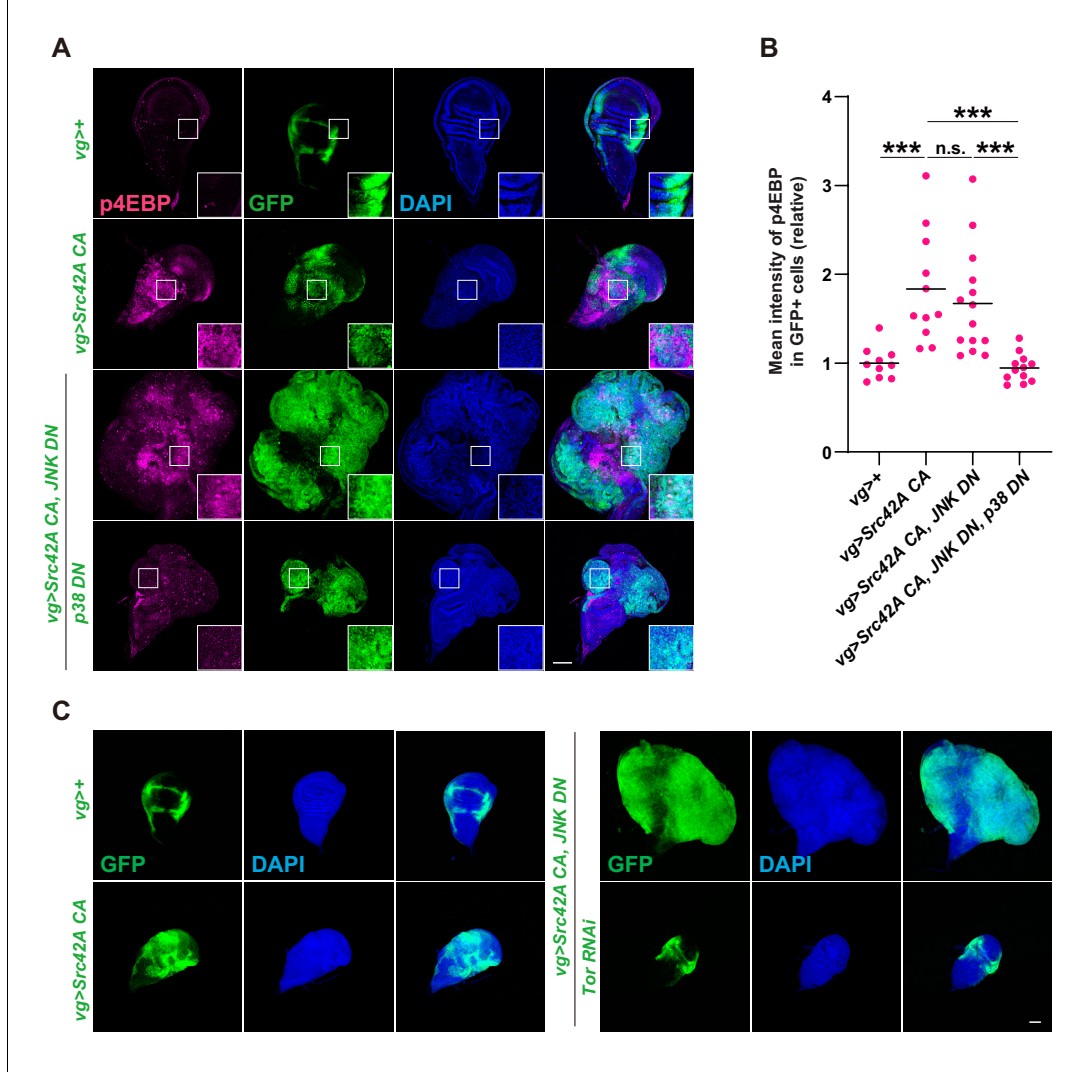

**Figure 5.** Tor signaling functions downstream of the Src-p38 pathway. (**A**) Src42A constitutively active (CA) induces phosphorylation of 4EBP, a readout of Tor activation. Src42A CA-mediated phosphorylation of 4EBP was suppressed by p38 inhibition but not by JNK inhibition. (**B**) Quantification of phosphorylated 4EBP in A. One-way ANOVA with Sidak's post-test. (**C**) Knockdown of *Tor* suppresses Src42A CA-induced cell proliferation. Scale bars, 100 μm.

The online version of this article includes the following figure supplement(s) for figure 5:

**Figure supplement 1.** Inhibition of Mef2 or Atf2 does not suppress Src42A constitutively active (CA)-induced cell proliferation.

activation and JNK inhibition (*Figure 6A–D*). Yeast dilution did not affect Src-induced p38 activation (*Figure 6—figure supplement 1A–B*), suggesting that nutrition signaling does not function upstream of p38. Importantly, yeast dilution increased survival over the Src-induced stress (*Figure 6E*), demonstrating the connection between organismal physiology and tumorigenesis.

Since yeast contains large amounts of amino acids, which activate Tor signaling (*Kim and Guan, 2019*; *Sabatini, 2017*), we investigated which amino acid might be involved in transformation by Src. To narrow down candidate amino acids, we compared the effects of essential amino acids and non-essential amino acids on survival over Src-induced stress. Feeding of essential amino acids, but not non-essential ones, enhanced lethality induced by *Src* expression (*Figure 7A*). We further narrowed down the responsible amino acid by repeating the essential amino acid feeding experiment, but this time subtracting each essential amino acid from the mixture one by one (*Figure 7B*). Out of all the essential amino acids, only methionine subtraction reversed lethality of the Src-activated flies induced by feeding essential amino acids (*Figure 7B*). This methionine specificity was unexpected

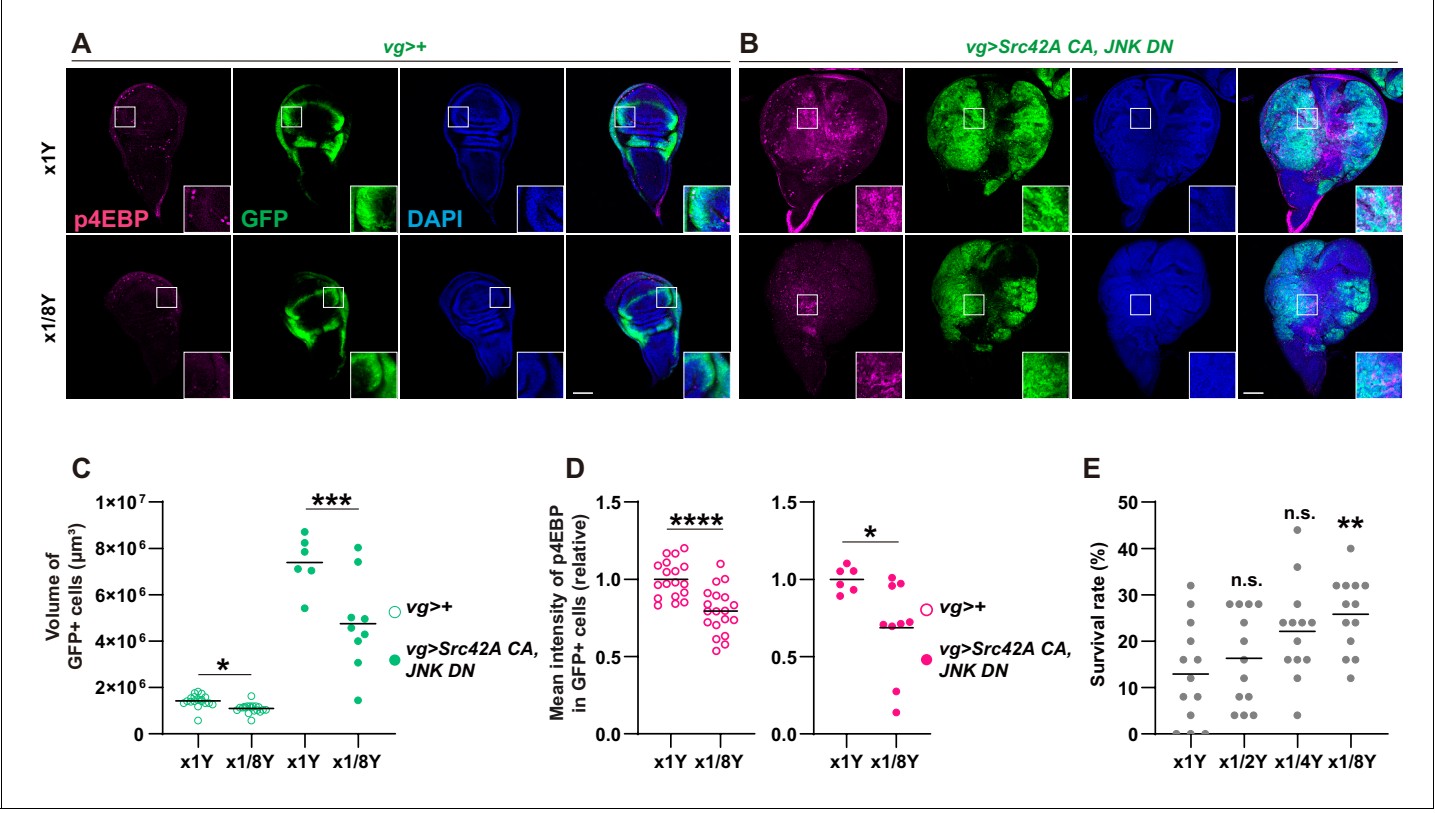

**Figure 6.** Yeast dilution affects Src-induced Tor signaling, tissue growth, and organismal lethality. (**A-B**) Dietary restriction of yeast suppresses both Src42A constitutively active (CA)-induced proliferation and phosphorylation of 4EBP. (**C**) Quantification of the total volume of GFP+ cells (μm³) in A-B. Mann-Whitney test. (**D**) Quantification of phosphorylated 4EBP in A-B. Two-tailed unpaired t-test. (**E**) Dietary restriction of yeast reduces organismal lethality caused by *Src42A CA* expression in the wing disc in a dose-dependent manner. One-way ANOVA with Sidak's post-test.

The online version of this article includes the following figure supplement(s) for figure 6:

**Figure supplement 1.** Yeast dilution does not affect Src-induced p38 phosphorylation.

because a variety of dietary amino acids, including serine, glycine, histidine, asparagine, cysteine, and methionine, have been shown to affect cancer outcome in mammals (*Badgley et al., 2020*; *Gao et al., 2019*; *Kanarek et al., 2018*; *Knott et al., 2018*; *Maddocks et al., 2013*; *Maddocks et al., 2017*). We found that methionine addition reduces the survival over the Src-induced stress (*Figure 7C*), whereas it does not affect survival of control animals (*Figure 7—figure supplement 1A*). Even in a detrimental situation where apoptosis is inhibited by JNK DN, methionine manipulation could affect the organismal survival over the Src-induced stress (*Figure 7—figure supplement 1B*).

We also investigated whether Src-induced tumor can cause a systemic effect on the nutritional state at the organismal level. Since tumor burden is clinically known to affect the amino acid profiles in the blood (*Lai et al., 2005*), we performed metabolomics analysis of the hemolymph, the fly blood, from the flies that bear Src-induced tumor in the wing discs. This metabolomics analysis revealed that the methionine concentration in the hemolymph is significantly decreased in flies that bear Src-induced tumors compared to control flies (*Figure 7D* and *Figure 7—figure supplement 1C–D*), which is correlated with the effects of dietary manipulation of methionine.

The correlative data on methionine in the amino acid subtraction experiment and the hemolymph analysis prompted us to investigate the role of methionine in Src-mediated oncogenesis. Feeding methionine enhanced Src-induced Tor signaling and overgrowth (*Figure 7E–H*), indicating that methionine is at least partly responsible for the nutrition-mediated activation of Tor signaling in Src-mediated tumors. Methionine addition did not affect Src-induced caspase activation (*Figure 7—figure supplement 2A*). Requirement of methionine for cell proliferation was relatively specific to

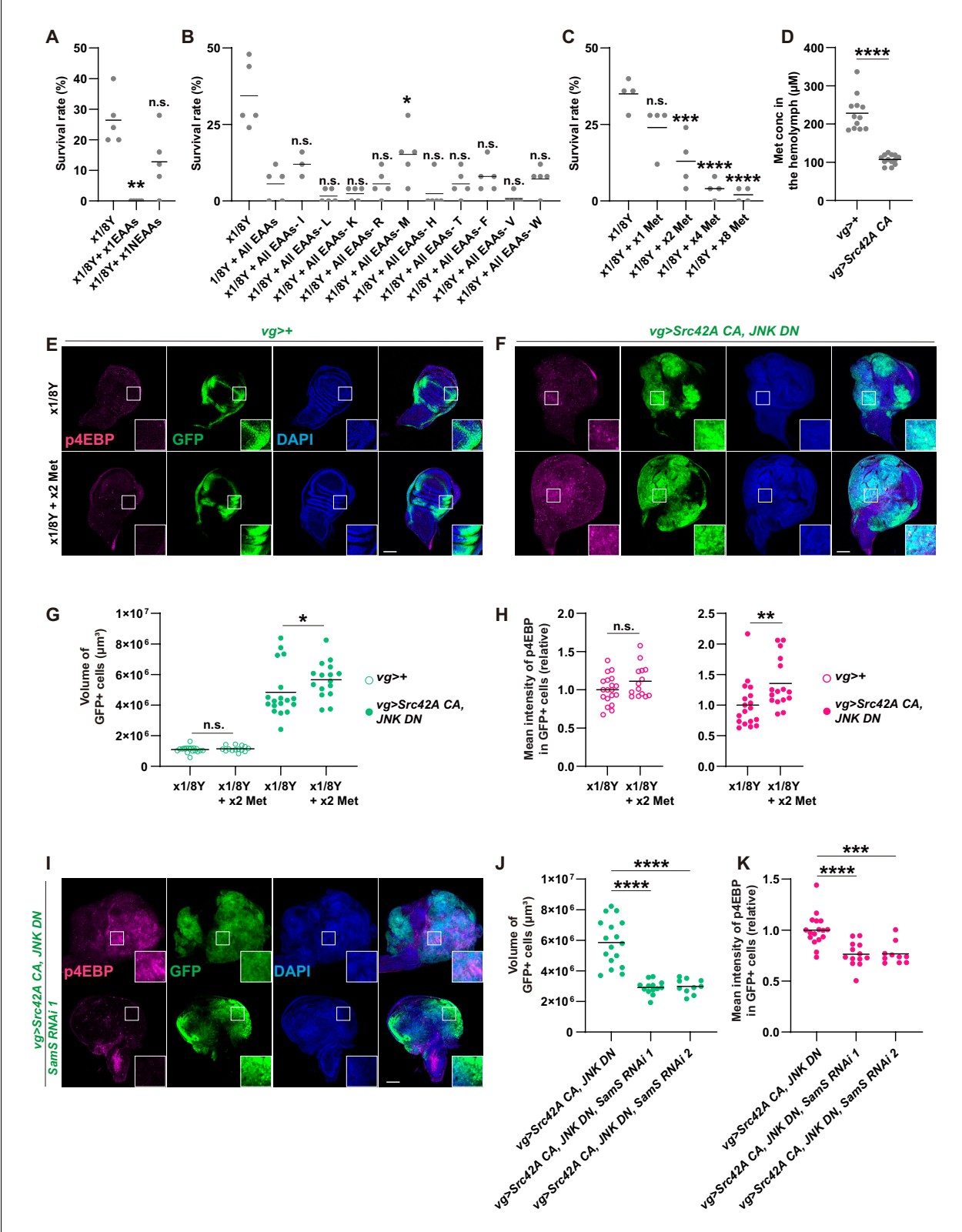

**Figure 7.** Methionine regulates Src-induced Tor signaling, tissue growth, and organismal lethality. (**A**) Addition of essential amino acids enhances organismal lethality caused by *Src42A constitutively active* (*CA*) expression in the wing disc, whereas addition of non-essential amino acids does not. Kruskal-Wallis test with Dunn's post-test. (**B**) Only methionine subtraction from the diet improves organismal survival over the Src42A CA stress. One-way ANOVA with Sidak's post-test. (**C**) Addition of methionine reduces organismal survival over the Src42A CA stress in a dose-dependent manner. *Figure 7 continued on next page*

*Figure 7 continued*

One-way ANOVA with Sidak's post-test. (**D**) An amount of methionine in the hemolymph was measured by LC-MS/MS. Expression of *Src42A CA* in the wing disc decreases the circulating methionine in the hemolymph. Two-tailed unpaired t-test. (**E-F**) Dietary methionine activates both cell proliferation and phosphorylation of 4EBP that are induced by Src42A CA and JNK DN. (**G**) Quantification of the total volume of GFP+ cells ($\mu m^3$) in E-F. Mann-Whitney test. (**H**) Quantification of phosphorylated 4EBP in E-F. Two-tailed unpaired t-test. (**I**) *SamS* knockdown suppresses phosphorylation of 4EBP and overgrowth induced by Src42A CA and JNK DN. (**J**) Quantification of the total volume of GFP+ cells ($\mu m^3$) in I. One-way ANOVA with Sidak's post-test. (**K**) Quantification of phosphorylated 4EBP in I. One-way ANOVA with Sidak's post-test. Scale bars, 100 $\mu m$.

The online version of this article includes the following figure supplement(s) for figure 7:

**Figure supplement 1.** Involvement of amino acids in Src-induced oncogenic stress.
**Figure supplement 2.** Involvement of methionine in Src-induced Tor activation.

tumor growth, because, under a normal condition without tumor, addition of methionine could not reverse the effects of yeast dilution, which induces smaller adult wings in control (*Figure 7—figure supplement 2B–C*).

Methionine can activate Tor through its conversion to SAM, a universal methyl donor (*Gu et al., 2017*). Consistently, inhibition of Sam synthetase, which converts methionine to SAM (*Obata and Miura, 2015*), suppressed both Src-induced overgrowth and Tor activation (*Figure 7I–K*). This indicates that methionine regulates Tor signaling in Src tumors in a tissue-autonomous manner. We also investigated a role for Samtor in Src-mediated oncogenesis. Samtor is a recently identified SAM sensor that inhibits Tor, which is released by SAM, in both mammals and *Drosophila* (*Gu et al., 2017*). As expected, in a diluted yeast condition, which suppresses Tor activation, *samtor* knockdown enhanced Tor activation with Src (*Figure 7—figure supplement 2D–F*). Unexpectedly, however, *samtor* knockdown suppressed Src-mediated overgrowth in spite of its Tor activation. We speculate that this growth suppression is likely attributed to Samtor's predicted methyltransferase function, which may not be related to Tor signaling. All together, these data indicate that methionine mediates Tor activation during Src-induced oncogenesis.

Finally, we explored a potential cross talk between Src-p38-Tor signaling and methionine-mediated Tor regulation. First, we investigated whether Src can activate methionine metabolism directly. Since *Src* expression in the wing disc lowers the methionine concentration in the hemolymph (*Figure 7D*), we hypothesized that Src tumors may uptake more methionine. Consistent with this idea, Src tumors uptake more methionine than control, which was demonstrated by performing an *in vitro* culture of the wing disc with a methionine analog homopropargylglycine (HPG) (*Figure 8A–B*). Furthermore, Src tumors exhibit a higher ratio of SAM to methionine (*Figure 8C*), indicating enhanced methionine flux to SAM. Collectively, Src enhances methionine uptake and methionine flux. Interestingly, this Src-mediated control of methionine metabolism was independent of p38 (*Figure 8D–F*, *Figure 8—figure supplement 1A*), suggesting that Src does activate methionine metabolism but not through p38.

## Discussion

Here, we elucidated the mechanism by which Src drives cell proliferation and cell death in an obligatory coupled manner. The obligation is mediated by coupling of two MAPK pathways diverging from the lynchpin protein Slpr. Downstream of Slpr, JNK activates cell death signaling, while p38 activates cell proliferation in a methionine-Tor dependent manner (*Figure 8G*). Src can potentially regulate Tor signaling through both p38-dependent and -independent mechanisms (*Figure 8—figure supplement 1B*). Our work provides several new insights discussed below.

First, our findings that Slpr mediates Src signaling provide a new molecular insight into regulation of Src signaling. *Drosophila* Src has been known to regulate various signaling pathways, including Notch, MAPKs, Jak-Stat, EGF, Wnt, and Hippo signaling (*Cordero et al., 2014*; *Enomoto and Igaki, 2013*; *Ho et al., 2015*; *Read et al., 2004*; *Tateno et al., 2000*), but Slpr has not previously been implicated in Src signaling. Especially, the mechanism behind Src-mediated JNK activation was elusive in spite of its biological importance in various contexts. Slpr fills in the gap between Src and JNK. In hindsight, it may seem sensible that Slpr, a JNKKK, could link Src and JNK. However, previous studies proposed that ubiquitin E2 complex Bendless (*Ma et al., 2013*) and F-actin cytoskeleton (*Enomoto and Igaki, 2013*; *Fernández et al., 2014*; *Rudrapatna et al., 2014*) mediate Src-JNK

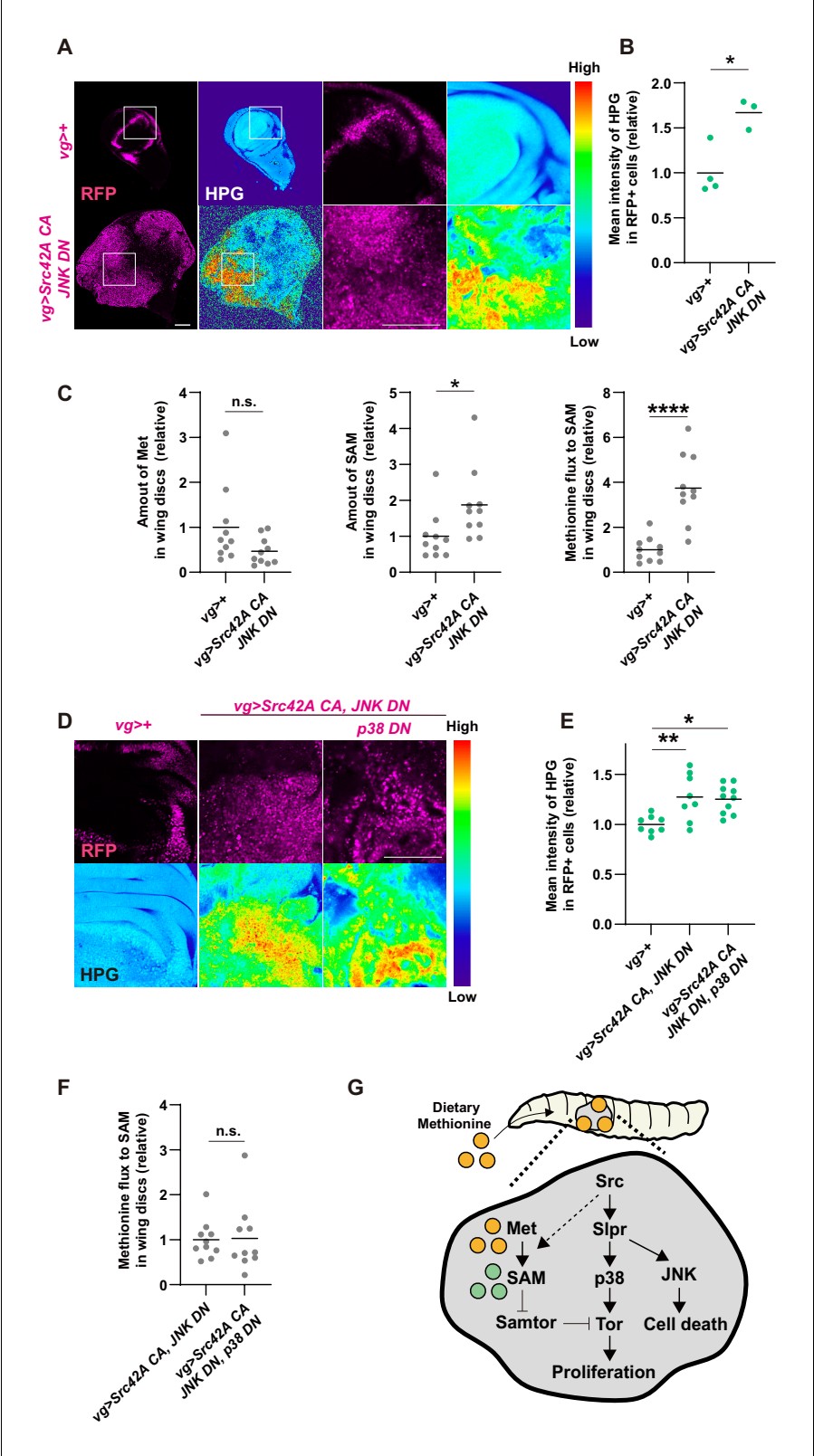

**Figure 8.** Cross talk between Src signaling and methionine-Tor signaling. (**A**) An *in vitro* culture of the wing disc with a methionine analog homopropargylglycine (HPG) demonstrates that the tumor disc induced by Src42A constitutively active (CA) and JNK DN uptakes more methionine than the control disc. (**B**) Quantification of the HPG intensity in A. Two-tailed unpaired t-test. (**C**) The amounts of methionine and SAM in the wing discs were measured by LC-MS. The tumor disc induced by Src42A CA and JNK DN contains a higher amount of SAM, whereas the amount of methionine is

*Figure 8 continued on next page*

*Figure 8 continued*

not significantly different. Methionine flux was calculated as a ratio of SAM and methionine. Two-tailed unpaired t-test. (D) The increase of methionine incorporation in the Src tumor is not mediated by p38. (E) Quantification of the HPG intensity in D. One-way ANOVA with Sidak's post-test. (F) p38 inhibition does not suppress the upregulated methionine flux by Src42A CA and JNK DN. Two-tailed unpaired t-test. (G) A schematic of the Src42A CA-mediated coupling of cell proliferation and death. JNK activates cell death, while p38 activates cell proliferation, which is regulated by methionine-mediated Tor signaling. Scale bars, 100 μm.

The online version of this article includes the following figure supplement(s) for figure 8:

**Figure supplement 1.** Cross-talk between Src signaling and Tor signaling.

signaling. Thus, it was unclear until now whether a MAPKKK is necessary for Src-mediated activation of JNK. Furthermore, there are five *Drosophila* JNKKKs, including dTAK1, Mekk1, Ask1, Wnd, and Slpr (*Nihalani et al., 2001*; *Ryabinina et al., 2006*; *Stronach and Perrimon, 2002*; *Takatsu et al., 2000*; *Wassarman et al., 1996*), each of which functions uniquely in a context-dependent manner. In our initial RNAi screening that identified Slpr as a Src effector, other MAPKKKs were not identified (*Figure 2—figure supplement 1A* and *Supplementary file 1*). Thus, identification of Slpr as a linker between Src and JNK provides a new insight. An urging, next question is how Src regulates Slpr. We speculate that the components that are considered as Src downstream and/or Slpr; upstream, such as Dok, Shark, and Misshapen (*Ríos-Barrera and Riesgo-Escovar, 2013*), may mediate the signal transduction between them. Interestingly, we also found that Slpr inhibition suppresses the phenotype of CA *Ras* overexpression (*Figure 8—figure supplement 1C, D*), which, similar to Src, simultaneously induces apoptosis and proliferation (*Karim and Rubin, 1998*). This suggests that Slpr could function as a lynchpin hub that integrates inputs from multiple oncogenes.

In this study, we exclusively focused on cell autonomous signaling induced by Src. But we noticed that Src elicits non-cell autonomous activation of MAPKs, cell death, and proliferation (*Figure 4A–B*, *Figure 4—figure supplement 1A–G*), which is reminiscent of the non-cell autonomous activation of Yorkie by Src (*Enomoto and Igaki, 2013*). It will be interesting to elucidate how non-cell autonomous signaling is regulated by Src activation in a future study.

Second, although Src was known to induce apoptosis as well as cell proliferation, how Src accomplishes this was unclear. We elucidated that, diverging from Slpr, p38 accelerates cell proliferation and that JNK induces cell death. This is an obligatory coupling of proliferation and death, likely being accomplished through evolution as an imperative mechanism to prevent tumorigenesis by a single oncogene activation. This type of fail-safe mechanism to prevent facile transformation was previously suggested in a context of *Myc* oncogene. We propose that, although each oncogene should have its unique fail-safe mechanism, the concept of the intrinsic fail-safe mechanism to prevent oncogenesis by a single oncogene is general.

Third, from a therapeutic perspective, our observation that methionine strongly regulates Src-mediated overgrowth is intriguing. Tumor growth *in vitro* is metabolically regulated by nutrition (*DeBerardinis and Chandel, 2016*; *Pavlova and Thompson, 2016*) and dietary manipulation of serine, glycine, histidine, asparagine, cysteine, or methionine could clinically modulate cancer outcome (*Badgley et al., 2020*; *Gao et al., 2019*; *Kanarek et al., 2018*; *Knott et al., 2018*; *Maddocks et al., 2013*; *Maddocks et al., 2017*). Notably, in our physiological *in vivo* condition, only subtraction of methionine from diet enhances organismal survival over Src-mediated oncogenic stress. Methionine has been studied in contexts of life span, metabolic health, and cancer together with other amino acids (*Gao et al., 2019*; *Lee et al., 2014*; *Malloy et al., 2013*; *Obata and Miura, 2015*; *Orentreich et al., 1993*; *Sanderson et al., 2019*), but the molecular mechanisms behind methionine-mediated cellular and organismal physiology were often unclear. We demonstrate that methionine regulates Tor activation, which controls cell proliferation induced by Src-p38 signaling.

In this study, we also found that the methionine concentration in the hemolymph is lower in flies that bear tumors in the wing disc, which is reminiscent of the clinical condition where tumor affects the amino acid profiles in the blood (*Lai et al., 2005*). Of note, local glutamine is known to be consumed in the tumor environment (*Sanderson et al., 2019*), but at least we did not observe reduction of glutamine in the hemolymph of the flies bearing tumors. We presume that Src-induced increase of methionine uptake in the Src tumor is at least partly responsible for the Src tumor-induced

hypomethioninemia, although other tissues may also contribute to it as the case of the fat body during wing disc repair (*Kashio et al., 2016*).

Regarding a cross-talk between Src signaling and nutrition-mediated Tor activation, we found that there are multiple cross-talk points. Src regulates methionine uptake and methionine flux in a p38-independent manner, both of which can potentially feed into Tor activation. Then, a question is how Src-p38 regulates Tor signaling, since Src-p38 clearly activates Tor signaling (*Figure 5A–B*). Although p38 is known to regulate Tor, its exact molecular mechanism remains unclear (*Cully et al., 2010*). Using the previously published RNAseq data on Src tumor in the wing disc (*Ho et al., 2015*), we surveyed expression levels of potential Tor regulators and selected genes that are affected by *Src* expression, including amino acid transporters and GATOR complexes. GATOR complexes regulate Tor through Rag GTPases (*Kim and Guan, 2019*; *Sabatini, 2017*). We examined whether their expression is regulated by Src in a p38-dependent manner using RT-qPCR. We found that among the amino acid transporters and GATOR complex components examined, only *pathetic* (*path*), an SLC36 amino acid transporter that can transport multiple amino acids, was significantly induced by Src in a p38-dependent manner (*Figure 8—figure supplement 1E, F*). Since Path can mediate amino acids-mediated Tor activation (*Goberdhan et al., 2005*; *Newton et al., 2020*), we speculate that Src-p38 could regulate Tor potentially through Path-mediated uptake of non-methionine amino acids.

Our findings have significant implications in the field of cancer therapeutics. As described in Introduction, SFK inhibitors have been clinically unsuccessful in spite of SFKs' contribution to tumorigenesis and metastasis (*Gargalionis et al., 2014*; *Sousa-Victor and Jasper, 2014*). We expect that the new insights our study provides on the Src tumorigenesis may help pave the way to cancer treatment. Furthermore, our data imply that nutritional state and tumorigenesis are closely linked. We speculate that, in case of tumors with a high SFK activity, manipulation of dietary methionine may have a clinical benefit.

# Materials and methods

## Key resources table

| Reagent type (species) or resource | Designation | Source or reference | Identifiers | Additional information |
|---|---|---|---|---|
| Gene *Drosophila melanogaster* | ras | Flybase | FLYB: FBgn0003205 | NA |
| Gene *Drosophila melanogaster* | src42A | Flybase | FLYB: FBgn0264959 | NA |
| Gene *Drosophila melanogaster* | vg | Flybase | FLYB: FBgn0003975 | NA |
| Gene *Drosophila melanogaster* | slpr | Flybase | FLYB: FBgn0030018 | NA |
| Gene *Drosophila melanogaster* | rpr | Flybase | FLYB: FBgn0011706 | NA |
| Gene *Drosophila melanogaster* | grim | Flybase | FLYB: FBgn0015946 | NA |
| Gene *Drosophila melanogaster* | hid | Flybase | FLYB: FBgn0003997 | NA |
| Gene *Drosophila melanogaster* | bsk | Flybase | FLYB: FBgn0000229 | NA |

*Continued on next page*

*Continued*

| Reagent type (species) or resource | Designation | Source or reference | Identifiers | Additional information |
|---|---|---|---|---|
| Gene *Drosophila melanogaster* | lic | Flybase | FLYB: FBgn0261524 | NA |
| Gene *Drosophila melanogaster* | tor | Flybase | FLYB: FBgn0021796 | NA |
| Gene *Drosophila melanogaster* | rl | Flybase | FLYB: FBgn0003256 | NA |
| Gene *Drosophila melanogaster* | p38b | Flybase | FLYB: FBgn0024846 | NA |
| Gene *Drosophila melanogaster* | mef2 | Flybase | FLYB: FBgn0003256 | NA |
| Gene *Drosophila melanogaster* | atf2 | Flybase | FLYB: FBgn0265193 | NA |
| Gene *Drosophila melanogaster* | SamS | Flybase | FLYB: FBgn0005278 | NA |
| Gene *Drosophila melanogaster* | samtor | Flybase | FLYB: FBgn0035035 | NA |
| Gene *Drosophila melanogaster* | CG13248 | Flybase | FLYB: FBgn0036984 | NA |
| Gene *Drosophila melanogaster* | tadr | Flybase | FLYB: FBgn0036984 | NA |
| Gene *Drosophila melanogaster* | CG9413 | Flybase | FLYB: FBgn0030574 | NA |
| Gene *Drosophila melanogaster* | jhl-21 | Flybase | FLYB: FBgn0028425 | NA |
| Gene *Drosophila melanogaster* | sbm | Flybase | FLYB: FBgn0030574 | NA |
| Gene *Drosophila melanogaster* | CG8757 | Flybase | FLYB: FBgn0036380 | NA |
| Gene *Drosophila melanogaster* | CG16700 | Flybase | FLYB: FBgn0030816 | NA |
| Gene *Drosophila melanogaster* | path | Flybase | FLYB: FBgn0036007 | NA |
| Gene *Drosophila melanogaster* | nprl3 | Flybase | FLYB: FBgn0036397 | NA |
| Gene *Drosophila melanogaster* | nprl2 | Flybase | FLYB: FBgn0030800 | NA |

*Continued on next page*

*Continued*

| Reagent type (species) or resource | Designation | Source or reference | Identifiers | Additional information |
|---|---|---|---|---|
| Gene *Drosophila melanogaster* | iml1 | Flybase | FLYB: FBgn0035227 | NA |
| Gene *Drosophila melanogaster* | wdr24 | Flybase | FLYB: FBgn0027518 | NA |
| Gene *Drosophila melanogaster* | wdr59 | Flybase | FLYB: FBgn0032339 | NA |
| Gene *Drosophila melanogaster* | nup44A | Flybase | FLYB: FBgn0033247 | NA |
| Gene *Drosophila melanogaster* | mio | Flybase | FLYB: FBgn0031399 | NA |
| Gene *Drosophila melanogaster* | RpL32 | Flybase | FLYB: FBgn0002626 | NA |
| Genetic reagent (*Drosophila melanogaster*) | UAS-RasV12 | Iswar Hariharan lab | UAS-RasV12 | NA |
| Genetic reagent (*Drosophila melanogaster*) | UAS-Src42A CA | Bloomington *Drosophila* Stock Center | BDSC: 6410 RRID:BDSC_6410 | NA |
| Genetic reagent (*Drosophila melanogaster*) | UAS-Src42A | Tian Xu lab | UAS-Src42A | NA |
| Genetic reagent (*Drosophila melanogaster*) | vg-Gal4 | Bloomington *Drosophila* Stock Center | BDSC: 6819 RRID:BDSC_6819 | NA |
| Genetic reagent (*Drosophila melanogaster*) | gmr-Gal4 | Iswar Hariharan lab | gmr-Gal4 | NA |
| Genetic reagent (*Drosophila melanogaster*) | TRE-RFP | Bloomington *Drosophila* Stock Center | BDSC: 59011 | NA |
| Genetic reagent (*Drosophila melanogaster*) | UAS-slpr RNAi | Bloomington *Drosophila* Stock Center | BDSC: 32948 RRID:BDSC_32948 | NA |
| Genetic reagent (*Drosophila melanogaster*) | UAS-slpr RNAi | Vienna *Drosophila* Resource Center | VDRC ID: 33516 RRID:FlyBase_FBst0460140 | NA |
| Genetic reagent (*Drosophila melanogaster*) | UAS-GFP | Iswar Hariharan lab | UAS-GFP | NA |
| Genetic reagent (*Drosophila melanogaster*) | UAS-his2B RFP | Iswar Hariharan lab | UAS-his2B RFP | NA |
| Genetic reagent (*Drosophila melanogaster*) | UAS-miRNA RGH | *Siegrist et al., 2010* | PMID:20346676 | NA |
| Genetic reagent (*Drosophila melanogaster*) | UAS-JNK DN | Iswar Hariharan lab | UAS-JNK DN | NA |

*Continued on next page*

*Continued*

| Reagent type (species) or resource | Designation | Source or reference | Identifiers | Additional information |
|---|---|---|---|---|
| Genetic reagent (*Drosophila melanogaster*) | UAS-p38 DN | Bloomington *Drosophila* Stock Center | BDSC: 59005 RRID:BDSC_59005 | NA |
| Genetic reagent (*Drosophila melanogaster*) | UAS-lic RNAi | Bloomington *Drosophila* Stock Center | BDSC: 31643 RRID:BDSC_31643 | NA |
| Genetic reagent (*Drosophila melanogaster*) | UAS-Tor RNAi | Bloomington *Drosophila* Stock Center | BDSC: 34639 RRID:BDSC_34639 | NA |
| Genetic reagent (*Drosophila melanogaster*) | UAS-erk RNAi 1 | Vienna *Drosophila* Resource Center | VDRC ID: 35641 RRID:FlyBase_FBst0461260 | NA |
| Genetic reagent (*Drosophila melanogaster*) | UAS-erk RNAi 2 | Vienna *Drosophila* Resource Center | VDRC ID: 109573 RRID:FlyBase_FBst0481239 | NA |
| Genetic reagent (*Drosophila melanogaster*) | UAS-mef2 RNAi | Vienna *Drosophila* Resource Center | VDRC ID: 15550 RRID:FlyBase_FBst0451917 | NA |
| Genetic reagent (*Drosophila melanogaster*) | UAS-atf2 RNAi | Bloomington *Drosophila* Stock Center | BDSC: 60124 RRID:BDSC_60124 | NA |
| Genetic reagent (*Drosophila melanogaster*) | UAS-SamS RNAi 1 | Vienna *Drosophila* Resource Center | VDRC ID: 7167 RRID:FlyBase_FBst0470579 | NA |
| Genetic reagent (*Drosophila melanogaster*) | UAS-SamS RNAi 2 | Vienna *Drosophila* Resource Center | VDRC ID: 103143 RRID:FlyBase_FBst0475005 | NA |
| Genetic reagent (*Drosophila melanogaster*) | UAS-samtor RNAi | Bloomington *Drosophila* Stock Center | BDSC: 54010 RRID:BDSC_54010 | NA |
| Genetic reagent (*Drosophila melanogaster*) | UAS-eiger | Iswar Hariharan lab | UAS-eiger | NA |
| Genetic reagent (*Drosophila melanogaster*) | UAS-rpr mts | Herman Steller lab | PMID:20837774 | NA |
| Genetic reagent (*Drosophila melanogaster*) | $w^{1118}$ | Erina Kuranaga lab | $w^{1118}$ | NA |
| Genetic reagent (*Drosophila melanogaster*) | Oregon R | Bloomington *Drosophila* Stock Center | BDSC: 4269 RRID:BDSC_4269 | NA |
| Antibody | Rabbit polyclonal phospho H3 antibody | Merck | Cat# 06–570 RRID:AB_310177 | Immunostaining (1:200) |
| Antibody | Rabbit polyclonal cleaved *Drosophila* Dcp-1 antibody | Cell Signaling | Cat# 9578 RRID:AB_2721060 | Immunostaining (1:100) |
| Antibody | Rabbit monoclonal phospho-p38 MAPK antibody | Cell Signaling | Cat# 4631 RRID:AB_331765 | Immunostaining (1:100) |
| Antibody | Mouse monoclonal phospho-Erk MAPK antibody | Merck | Cat# M8159 RRID:AB_477245 | Immunostaining (1:100) |

*Continued on next page*

*Continued*

| Reagent type (species) or resource | Designation | Source or reference | Identifiers | Additional information |
|---|---|---|---|---|
| Antibody | Rabbit monoclonal phospho-4EBP1 antibody | Cell Signaling | Cat# 2855 RRID:AB_560835 | Immunostaining (1:100) |
| Antibody | Rabbit polyclonal phospho-S6 antibody | *Kim et al., 2017* | PMID:28829944 | Immunostaining (1:300) |
| Antibody | Alexa mouse Fluor 568 secondary antibody | Thermo Fisher | Cat# A-11004 RRID:AB_253407 | Immunostaining (1:300) |
| Antibody | Alexa rabbit Fluor 568 secondary antibody | Thermo Fisher | A-11036 RRID:AB_10563566 | Immunostaining (1:300) |
| Sequence-based reagent | slpr (primer) | This paper | NA | F: 5'-CTACAAGGGCTTCGATCCGTTG-3 R: 5'-GTTTGCCAGCAGCTCTTCATCAG-3 |
| Sequence-based reagent | slpr (primer) | This paper | NA | F: 5'-CAATCATCTGCAGCAGAAGACGC-3' R: 5'-CATCGGAGAATTTGGAATAGGTGC-3' |
| Sequence-based reagent | SamS (primer) | *Obata and Miura, 2015* | PMID:32938923 | F: 5'-GCCAACGGCGTTCATATC-3' R: 5'-GGCATATCCAAACATGATACCC-3' |
| Sequence-based reagent | CG13248 (primer) | FlyPrimerBank | PP18106 | F: 5'-AAACCGATGCCTCAACACCTT-3' R: 5'-CAGTCAGCACGTAGATGCCA-3' |
| Sequence-based reagent | tadr (primer) | FlyPrimerBank | PP20579 | F: 5'-CAGCCCGCTGTAAAACTAGC-3' R: 5'-GGCCAGAGCATCTAGCCAG-3' |
| Sequence-based reagent | CG9413 (primer) | FlyPrimerBank | PP29104 | F: 5'-TGGGGTGGCTTTAATTGTTGG-3' R: 5'-CAGTGCGAACCAGTAAACCG-3' |
| Sequence-based reagent | jhl-21 (primer) | *Newton et al., 2020* | PMID:32938923 | F: 5'-TCAAGCGGAAGCTAACACTCA-3' R: 5'-TTCGGTGTAAATAAA-GACTCCCG-3' |
| Sequence-based reagent | sbm (primer) | FlyPrimerBank | PP3597 | F: 5'-AATGTGCCAACAAAAACAACGA-3' R: 5'-GTCCCTGATGAGTCGGTCTC-3' |
| Sequence-based reagent | CG8757 (primer) | *Newton et al., 2020* | PMID:32938923 | F: 5'-AGAAACGATTGGATCGGGCA-3' R: 5'-ATCTGCCATCTTTTGGACCGA-3' |
| Sequence-based reagent | CG16700 (primer) | FlyPrimerBank | PP25676 | F: 5'-CCTACAAGCTATCTGGAGACCA-3' R: 5'-GAGACCTCCGTTCTTGAAGGC-3' |
| Sequence-based reagent | path (primer) | *Newton et al., 2020* | PMID:32938923 | F: 5'-TGTTTGATTTGCGCGGCATT-3' R: 5'-TTCGACCCGCTGTCCACTAT-3' |
| Sequence-based reagent | nprl3 (primer) | FlyPrimerBank | PP28256 | F: 5'-GTTAAACCACAGCTATGCAACCA-3' R: 5'-CAGAGTGGGATGACTGACAAAG-3' |

*Continued on next page*

*Continued*

| Reagent type (species) or resource | Designation | Source or reference | Identifiers | Additional information |
|---|---|---|---|---|
| Sequence-based reagent | nprl2 (primer) | FlyPrimerBank | PP27923 | F: 5'-TTCAACGCTGCATTCTCACC-3'<br>R: 5'-ATTCCGTGCGTACTTCTGCTG-3' |
| Sequence-based reagent | iml1 (primer) | FlyPrimerBank | PP8389 | F: 5'-CGTGGCTGCAACAAATCCTAC-3'<br>R: 5'-GCCCGATTCTATGCTTATCACA-3' |
| Sequence-based reagent | wdr24 (primer) | FlyPrimerBank | PP3395 | F: 5'-GCCCTGGCCCTGAATAAGG-3'<br>R: 5'-TGAAGCCATTGCTGTTTATGGAG-3' |
| Sequence-based reagent | wdr59 (primer) | FlyPrimerBank | PP2977 | F: 5'-GCACCCGAACAAACGTACATC-3'<br>R: 5'-CCGAGTAATCAACC-GACATGG-3' |
| Sequence-based reagent | nup44A (primer) | FlyPrimerBank | PP28620 | F: 5'-GAGGAGGTGATTGGCGAAAAG-3''<br>R: 5'-GCGAGTCTACAAGGGTGGTG-3' |
| Sequence-based reagent | mio (primer) | NA | PMID:26024590 | F: 5'-AGCGAGACGAGCTAAACAATTC-3'<br>R: 5'-GTGTAAGAGGCAAG-CAAAGGTT-3' |
| Sequence-based reagent | RpL32 (primer) | This paper | NA | F: 5'-CCAGCATACAGGCCCAAGATCGTG-3'<br>R: 5'-TCTTGAATCCGGTGGGCAGCATG-3' |
| Commercial assay or kit | Methionine analog homopropargylglycine (HPG)- based on Click-iT HPG Alexa Fluor 488 Protein Synthesis Assay kit Single Cell 3' Library and Gel Bead Kit v2 | Invitrogen | C10428 | NA |
| Commercial assay or kit | Maxwell RSC simply RNA Tissue Kit | Promega | AS1340 | NA |
| Commercial assay or kit | ReverTra Ace qPCR RT Kit | Toyobo | FSQ-101 | NA |
| Commercial assay or kit | BCA protein assay kit | Thermo Fisher | 23225 | NA |
| Software, algorithm | ImageJ | NA | https://imagej.nih.gov/ij/ RRID:SCR_003070 | NA |
| Software, algorithm | IMARIS 9.5.1 | Oxford Instrument | https://imaris.oxinst.com/packages RRID:SCR_007370 | NA |
| Other | DAPI | Sigma | D9542 | 1:1000 |

### *Drosophila* husbandry

Flies were maintained as previously described (*Yoo et al., 2016*). The fly food is composed of the following ingredients: 0.8% agar, 10% glucose, 4.5% corn flour, 3.72% dry yeast, 0.4% propionic acid, 0.3% butyl *p*-hydroxybenzoate.

The composition of the food for yeast restriction and amino acid addition/subtraction is described in *Supplementary file 1* and *Supplementary file 2*. The amino acid concentrations are based on the previously described holidic medium for *Drosophila melanogaster* (*Piper et al., 2014*).

### *Drosophila* stocks

Flies were crossed and raised at 25°C unless otherwise noted. For wild-type controls, Oregon-R (Bloomington stock center [BL] 4269) was used. The following fly stocks were used in this study:

- *UAS-Src42A CA* (BL6410)
- *UAS-Src42A* (a gift from Dr Tian Xu) *vg-Gal4* (BL6819) *gmr-Gal4* (a gift from Dr Iswar Hariharan)
- *TRE-RFP* (a gift from Dr Dirk Bohmann)
- *UAS-slpr RNAi* (BL32948, v33516)
- *UAS-GFP* (a gift from Dr Iswar Hariharan)
- *UAS-his2B RFP* (a gift from Dr Iswar Hariharan)
- *UAS-miRNA RGH* (a gift from Dr Iswar Hariharan)
- *UAS-JNK DN* (a gift from Dr Iswar Hariharan)
- *UAS-p38 DN* (BL59005)
- *UAS-lic RNAi* (BL31643)
- *UAS-Tor RNAi* (BL34639)
- *UAS-erk RNAi* (RNAi 1: v35641, RNAi 2: v109573)
- *UAS-RasV12* (a gift from Dr Iswar Hariharan)
- *UAS-mef2 RNAi* (v15550)
- *UAS-atf2 RNAi* (BL60124)
- *UAS-SamS RNAi* (RNAi 1: v7167, RNAi 2: v103143)
- *UAS-samtor RNAi* (BL54010)
- *UAS-rpr^{mts}* (a gift from Hermann Steller) (*Sandu et al., 2010*)
- *UAS-eiger* (a gift from Dr Iswar Hariharan) *w^{1118}* (a gift from Dr Erina Kuranaga)

## Immunofluorescence and confocal imaging

Wing discs were dissected in PBS, fixed with paraformaldehyde in PBS, and washed in PBS with 0.1% Triton X-100. We used the following antibodies and fluorescent dyes: rabbit phospho H3 antibody (1:200, 06–570, Merck), rabbit cleaved *Drosophila* Dcp-1 antibody (1:100, #9578, Cell Signaling), rabbit phospho-p38 MAPK antibody (1:100, #4631, Cell Signaling), mouse phospho-Erk MAPK antibody (1:100, M8159, Merck), rabbit phosphor-4EBP one antibody (1:100, #2855, Cell Signaling), rabbit phospho-S6 antibody (1:300) (*Kim et al., 2017*), DAPI (D9542, Sigma), and Alexa Fluor secondary antibodies (A11004, A11008). Fluorescent images were acquired with confocal microscopes (Zeiss LSM 880,780). Quantification of the cell volume and the intensity measurement of fluorescent signals were performed by using IMARIS 9.5.1. ImageJ was used to measure the wing size.

## Immunostaining quantification

For quantification of the fluorescent intensity for TRE-RFP, p-p38, pERK, p4EBP, and pS6 in GFP+ cells, GFP+ cell volume and sum intensity of fluorescent signals in GFP+ cells were used. Samples were analyzed with IMARIS 9.5.1. The GFP intensity was measured to identify GFP+ cells. The volume and the sum intensity of fluorescent signals were quantified by surface function. The mean intensity within GFP+ cells was calculated by dividing the sum intensity of fluorescent signals by GFP + cell volume. For quantification of the fluorescent intensity in GFP- cells, the total disc size ($\mu m^3$) and the sum intensity of fluorescent signals within the whole disc were measured. DAPI intensity was used to determine the whole disc region. Volume of GFP- cells and sum intensity in GFP- cells were quantified by subtracting each information (volume, sum intensity) in GFP+ cells from ones in whole discs. The mean intensity of fluorescence within GFP- cells was calculated as performed in GFP+ cells.

## Quantification of proliferating cells

For quantification of proliferation rate, the number of pH3-positive cells and GFP+ cell area were measured by using ImageJ. The density of pH3+ cells was calculated by dividing the number of pH3 + cells by GFP+ cell area.

## Quantification of dying cells

To determine the dying cells, the intensity of cDCP1 antibody staining was used. Percentage of DCP1-positive cells was measured in GFP+/- cells respectively by using ImageJ.

## Measurement of survival rate

For measuring the survival rate, mated females were allowed to lay eggs on a grape agar plate for 24 hr at 25℃. First instar larvae were collected from the grape agar plate and placed into treatment vials with different food conditions. Each vial contains 50 larvae. The number of adult flies of each genotype that were able to eclose was recorded. Survival rates were calculated as the number of adult flies that eclosed divided by the expected number of larvae of each genotype placed in each vial. Most experiments were performed at 25℃, except the ones performed at 23℃ to increase the sensitivity of the assays in Figures 2A, 3E, 4F, 6E and 7A–C, *Figure 4—figure supplement 1H* and *Figure 7—figure supplement 1A–B*.

## Measurement of amino acids in the larval hemolymph

We teared 20 L3 larvae on an iced block to collect 5 µl of the hemolymph. Extraction and quantification of metabolites in the hemolymph were performed as described previously with capillary electrophoresis time-of-flight mass spectrometry (*Soga et al., 2006*; *Soga et al., 2003*; *Tabata et al., 2017*).

## Measurement of methionine and SAM in the larval wing discs and hemolymph by LC-MS/MS

Twenty wing discs or 1.5 µl hemolymph were used per sample to measure the amounts of methionine and/or SAM. Frozen samples in 1.5 ml plastic tubes were homogenized in 300 µl of cold methanol with 1× f3-mm zirconia beads using an automill (Tokken Inc) at 41.6 Hz for 2 min. The homogenates were mixed with 200 µl of methanol, 200 µl of $H_2O$, and 200 µl of $CHCl_3$ and then vortexed for 20 min at RT. The samples were centrifuged at 20,000 g for 15 min at 4℃. The supernatant was mixed with 350 µl of $H_2O$ and vortexed for 10 min at RT. The aqueous phase was collected after centrifugation and dried in a vacuum concentrator. The samples were redissolved by 50% acetonitrile, transferred to autosampler vial, and kept at 5.0℃. The insoluble pellets were heat-denatured with 0.2 N NaOH and used to quantify total protein using a BCA protein assay kit (Thermo). Chromatographic separations in an Acquity UPLC H-Class System (Waters) were carried out under reverse-phase conditions using an Acquity UPLC HSS T3 column (2.1 × 100 mm) in infusion. A mobile phase consists of solvent A (10 mM ammonium bicarbonate, pH 7.5) and solvent B (acetonitrile). The flow rate was 0.5 ml/min at 30.0℃. Compounds were separated by gradient elution, in turn ionized and detected using a Xevo TQ-S triple quadrupole mass spectrometer coupled with an electro-spray ionization source (Waters). Precursor ion was scanned at m/z (MH+: 399.143 > 250.092 for SAM, and 150.057 > 104.053 for methionine) by multiple reaction monitoring and established methods using individual authentic compounds and biological samples. The peak area of a target metabolite was analyzed using MassLynx 4.1 software (Waters). Metabolite signals were then normalized to the total protein level of the corresponding sample after subtracting the values from the blank sample. A two-tailed unpaired t-test was used to test between samples.

## RT-qPCR

For RT-qPCR, the total RNA was extracted from 30 discs per sample by using the Maxwell RSC simplyRNA Tissue Kit (Promega). Total RNA (250 ng) was subjected to DNase digestion, followed by reverse transcription using the ReverTra Ace qPCR RT Kit (Toyobo). qPCR was performed using the FastStart Essential DNA Green Master Mix (Roche). Rpl32 was used as an internal control. Error bars represent the SE. Primers used for qPCR are shown in *Supplementary file 4*.

## HPG incorporation assay

Methionine incorporation was monitored by the methionine analog HPG-based on Click-iT HPG Alexa Fluor Protein Synthesis Assay kit (Invitrogen). Wing discs were dissected in S2 medium. After dissection, discs were incubated with the S2 medium containing 5 mM HPG for 30 min. After washing with PBS, the discs were fixed by 4% PFA for 30 min. The discs were then washed by PBS three

times and permeabilized with PBS with 0.1% Triton X-100 for 20 min, followed by 15 min incubation with freshly prepared Click-iT reaction cocktail. The discs were then washed by PBS three times and PBS with 0.1% Triton X-100 once, and incubated with DAPI for 30 min. Fluorescent images were acquired with the confocal microscope (Zeiss LSM 880). The mean intensity of HPG in *Src42A CA* expressing cells were quantified by using ImageJ.

## Statistical analysis

Statistical tests used were indicated in the figure captions. All the data plotted in each graph were summarized in *Supplementary file 3*. Sample sizes were determined empirically based on the observed effects. All the statistical analyses were performed using Graphpad Prism 9. A two-tailed unpaired t-test was used to test between two samples. The number of samples is shown in *Supplementary file 3*. One-way ANOVA with Sidak's multiple comparisons test was used to test among groups. Statistical significance is shown by asterisk; $*p<0.05$, $**p\leqq0.01$, $***p\leqq0.001$, $****p\leqq0.0001$.

## Acknowledgements

This work was supported by AMED-PRIME (17939907) and the JSPS KAKENHI (JP16H0622) to SKY and by National Research Foundation of Korea grant (NRF-2020R1A5A1018081) to JC.

# Additional information

### Funding

| Funder | Grant reference number | Author |
|---|---|---|
| Japan Society for the Promotion of Science | JP16H0622 | Sa Kan Yoo |
| Japan Agency for Medical Research and Development | 17939907 | Sa Kan Yoo |
| National Research Foundation of Korea | NRF-2020R1A5A1018081 | Jongkyeong Chung |

The funders had no role in study design, data collection and interpretation, or the decision to submit the work for publication.

### Author contributions

Hiroshi Nishida, Conceptualization, Formal analysis, Investigation, Writing - review and editing; Morihiro Okada, Lynna Yang, Sho Tabata, Tomoyoshi Soga, Investigation, Methodology; Tomomi Takano, Formal analysis, Investigation, Methodology; Diana M Ho, Resources, Formal analysis; Jongkyeong Chung, Resources, Methodology; Yasuhiro Minami, Supervision, Project administration, Writing - review and editing; Sa Kan Yoo, Conceptualization, Formal analysis, Supervision, Funding acquisition, Investigation, Writing - original draft

### Author ORCIDs

Hiroshi Nishida https://orcid.org/0000-0003-0284-5771
Jongkyeong Chung http://orcid.org/0000-0001-5894-7537
Sa Kan Yoo https://orcid.org/0000-0003-3358-4818

### Decision letter and Author response

Decision letter https://doi.org/10.7554/eLife.59809.sa1
Author response https://doi.org/10.7554/eLife.59809.sa2

## Additional files

### Supplementary files

- Supplementary file 1. Stock information and results in the first screening. The table shows stock IDs and results of survivor numbers and the wing phenotype.

- Supplementary file 2. Food composition. The table shows components of the fly foods that were used for dietary restriction, amino acid subtraction, and methionine addition experiments.

- Supplementary file 3. All the statistics data. The table shows all the plotted data and information of statistic analyses performed in this manuscript.

- Supplementary file 4. Primer information. The table shows sequences of the primers used in this study.

- Transparent reporting form

### Data availability

All data generated or analysed during this study are included in the manuscript and supporting files.

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
