## [Decision Letter]

**Acceptance summary:**

In this manuscript, the authors address some of the most critical issues regarding the function of upstream components of JNK signaling that have lingered around, unsolved, for quite some time. Although it was known that a key activation point was at the level of a KKKinase, which was almost certain to be slipper, the question of how this pathway controls multiple functions including the apparently contradictory phenotypes of proliferation and cell death has been worked out in some detail in this paper and the study opens, as well, doors for further analysis of how a cell keeps information channelized while also allowing for the possibility of cross talk that maintain homeostatic balance.

**Decision letter after peer review:**

Thank you for submitting your article "Methionine restriction breaks obligatory coupling of cell proliferation and death by an oncogene Src in *Drosophila*" for consideration by *eLife*. Your article has been reviewed by 3 peer reviewers, one of whom is a member of our Board of Reviewing Editors, and the evaluation has been overseen by Utpal Banerjee as the Senior Editor. The reviewers have opted to remain anonymous.

The reviewers have discussed the reviews with one another and the Reviewing Editor has drafted this decision to help you prepare a revised submission. Normally, we summarize the reviews at this point to list the specific list of experiments that we expect the authors to conduct. Given the difficulties these days with performing extensive experiments, we provide you with the full reviews so that you can determine the critical issues on which you should focus when carrying out revisions. You will see that the src-slpr aspect is universally lauded as novel. The issues raised have to do with the methionine/TOR angle. It should be easy to determine from the reviews the aspects that are most important to the reviewers. Some parts are easily addressed. If you so wish, please feel free to let us know if there is some aspect that might be particularly difficult to handle experimentally, and I will run this by the reviewers. Please also note the following *eLife* policy:

The editors have judged that your manuscript is of interest to *eLife*, but as described below additional experiments are required before it is published. We would like to draw your attention to changes in our revision policy that we have made in response to COVID-19 (https://elifesciences.org/articles/57162).

First, because many researchers have temporarily lost access to the labs, we will give authors as much time as they need to submit revised manuscripts. We are also offering, if you choose, to post the manuscript to bioRxiv (if it is not already there) along with this decision letter and a formal designation that the manuscript is "in revision at *eLife*". Please let us know if you would like to pursue this option. (If your work is more suitable for medRxiv, you will need to post the preprint yourself, as the mechanisms for us to do so are still in development.)

Summary:

This study by Nishida et al. investigates the mechanisms by which the well-known phenomenon of constitutive activation of Src mediates both cell growth and apoptosis during tumorigenesis. In particular, the authors sought to explore whether these two pathways were coupled and through what downstream effectors Src functions. They found that indeed both cell growth and apoptosis were obligately coupled through the "lynchpin" protein Slpr, which mediates cell growth through a Src-p38-Tor pathway and apoptosis through a Src-JNK pathway.

Reviewer #1:

Overall, the authors the authors do a good job rationalizing, designing, explaining, and interpreting most of their results and the important contributions and insights of this paper in both identifying that Src mediates JNK activity, which was not previously understood well, and that it does this through through slpr provide important knowledge to the field. Although the genetic experiments are straightforward, the link with TOR and nutrition in interpreting the phenotype is more complex and difficult to explain as written.

1. The mechanistic link between Src-Slpr-p38 and TOR is not well developed. Although it is clear that TOR inhibition inhibits overgrowth, these experiments do not provide strong support for TOR being a bona fide component of the Src-Slpr-p38 axis. For example, loss of function of TOR is expected to have rather pleiotropic effects. Stronger mechanistic evidence of how p38 activates TOR, would greatly strengthen this line of reasoning. Currently, less direct mechanisms cannot be excluded.

2. The link between the genetic experiments and nutrition, for example the role of methionine, is quite interesting, but is not so well developed and the reasoning behind the experiments is difficult to follow. It seems as though the logic is that diluting amino acids decreases the tumor growth, and that adding the amino acids back suppresses the rescue. Addition of amino acid, without methionine, limits tumor growth to a degree and increases survival compared to EAAs, though the magnitude of the rescue is difficult to interpret (e.g. 7.5 % vs 15 % survival, with quite large variance). Adding methionine alone clearly limits survival of flies with tumor, though it is not clear that block is caused by TOR alone. These latter metabolic experiments are fairly difficult to interpret, and the link between Src-Slpr-p38 and TOR and methionine is not as clear as the work presented earlier in the paper.

3. Are there regional differences within the wing disc (Figure 1) where proliferation (pouch) vs apoptosis (edges) observed more commonly? How would such a difference fit into the model? Clonal analysis could shed light on this issue of autonomy

Reviewer #2:

In this paper, Nishida et al. address a long-standing and intensely-investigated question, namely how are cell proliferation and cell death controlled to modulate tissue growth in animals. They specifically investigate the effects of Src overexpression in imaginal tissues (which leads to both increases cell proliferation and cell death) and they make two key observations: (a) they identify the kinase Slpr as an effector of Src in the control of both cell proliferation (via p38) and cell death (via JNK); (b) they show that the effects of p38 are mediated via p38 and that dietary methionine can control TOR and modulate Src-mediated cell proliferation.

Overall, I liked this paper. I think the two main findings will be of interest to, not just *Drosophila* researchers, but also cell, developmental and cancer biologists interested in the regulation of tissue growth.

I have two main comments/questions:

1. Are the effects on cell proliferation and cell death strictly autonomous? It's hard to tell clearly from Figure 1, especially since the choice of driver of vg-Gal4 doesn't have clearly demarcated expression boundaries (compared to, say, en-GAL4 or ap-GAL4). This is import since, if the proliferation is a result of cell-death induced compensatory proliferation, then the interpretations of the data in the paper would be very different than if everything was strictly autonomous.

2. The link between p38 and TOR is interesting, but it relies on using measurement of phospho-4EBP as a readout of TOR signaling. These data are certainly very clean, however, it’s important to note that total levels of 4EPB are often induced upon stress signaling. Hence, it's important to be sure that the increase in phospho-4EBP signal does reflect an increase in the amounts of phosphorylated 4EBP (a TOR effect) vs simply an increase in total 4EBP (a TOR-independent, stress effect). A recent paper from the Karpac lab (Cell Reports, 2020) had some nice immunostaining with a non-phospho 4EBP, which would be a useful control in this manuscript. Alternatively, the induction of TOR could be confirmed with a phospho-S6k or phospho-S6 antibody (western on discs or immunostaining). The genetic expts with TOR certainly support a role for TOR in modulating Src function, but they could equally be interpreted as TOR signaling being a parallel pathway or TOR being a downstream Src pathway.

Reviewer #3:

The authors examine how an oncogene, Src, can activate both proliferation and apoptosis during oncogenesis. The answer is surprisingly simple. In the case of Src, it activates both pro-apoptotic (JNK) and proliferative (p38) MAPK pathways and upstream of this, the authors identify the MPKKK slpr as the lynchpin in this pathway. Src also activates Erk-type MAPK, but according to the authors, this has no effect for tumorigenesis. The authors then focus on the role of p38 and found that p38 controls Tor signaling. Interestingly, the nutrition status does influence Src-induced Tor signaling and tissue growth. To further delineate this observation, the authors tested each amino acid and found specifically a requirement of Methionine for Src-induced tumor growth. They also observed that the Methionine concentration in the hemolymph of these Src-bearing larvae is lower compared to controls, presumably because the Src-tumor tissue absorbs more of it. Overall, the authors present a model that combines several signaling pathways and metabolism for Src-induced tumor growth.

This is an interesting manuscript, but I do have some comments and suggestions for improvement.

1. What is the nature of Src42A CA? How does it promote dominant activity to Src?

2. Provide a complete picture of the Src CA suppressor screen in which they found slpr. Were there other suppressors and if so, why did they focus on slpr?

3. Present the results from the screen of the other MPKKK. Did they also screen the MPKK and other upstream components in the MPK pathway (Msn, Traf1, Traf2, Rho1)? This might tell us how Src activates Slpr.

4. How does Src activate Slpr? Is any of the domains of Slpr (SH2, SH3, kinase) required for this activation?

5. How many different slpr RNAi lines were tested? Are the results verified with several independent lines? Do slpr mutants suppress Src42ACA?

6. Verify the Erk-MAPK (rl) RNAi line(s).

7. The data presentation needs improvement. The discs are too small and often the relevant staining (PH3, cDcp1, TRE-RFP, etc.) is hard to judge. This is also particularly important for the non-cell autonomy of Erk and p38 activity in Figure 4A-B. Some convincing close-up pictures could help.

8. There is no comprehensive description of how the quantitation of the staining was conducted. Was the quantitation done in the region of overgrowth? How was the volume of GFP+ tissue determined? Were equivalently sized regions used and how was their level of fluorescence assessed? Was there some internal control to normalize for differences in general staining between regions of interest and region where no change would be expected (ie a part of the disc outside of the vg-Gal4 area)?

9. Figure 1A: quantify the cDcp1 staining.

10. Figure 2C: slpr RNAi seems to suppress TRE-RFP, but not overgrowth. In fact, the slpr RNAi disc (lower panels) is significantly larger than the src42ACA disc (middle panels) including the GFP+ area. That is inconsistent with the authors model. Explain. Please quantify GFP+ areas and overall disc size.

11. Figure 3B: Perform PH3 labeling in vg>Src42ACA JNKDN slprRNAi background. I would like to know if the overgrowth and PH3 labeling in Src42ACA JNKDN is dependent on Slpr. The authors imply this, but don't show it.

12. Figure 5A-B: is the 4EBP phosphorylation in vg>Src42CA JNK DN discs depended on Slpr, i.e. does slpr RNAi block it?

13. In the Discussion, the authors mention that "methionine regulates Tor activation". I don't think this direct link was established in this paper.

14. Is the a SAMTOR homolog in flies? If so, test it in the Src42ACA screen.

---

## [Author Response]

Reviewer #1:Overall, the authors the authors do a good job rationalizing, designing, explaining, and interpreting most of their results and the important contributions and insights of this paper in both identifying that Src mediates JNK activity, which was not previously understood well, and that it does this through through slpr provide important knowledge to the field. Although the genetic experiments are straightforward, the link with TOR and nutrition in interpreting the phenotype is more complex and difficult to explain as written.1. The mechanistic link between Src-Slpr-p38 and TOR is not well developed. Although it is clear that TOR inhibition inhibits overgrowth, these experiments do not provide strong support for TOR being a bona fide component of the Src-Slpr-p38 axis. For example, loss of function of TOR is expected to have rather pleiotropic effects. Stronger mechanistic evidence of how p38 activates TOR, would greatly strengthen this line of reasoning. Currently, less direct mechanisms cannot be excluded.

In the original manuscript, we had three pieces of data related to Tor signaling.

1. Src-p38 activates Tor signaling, based on p4EBP.

2. Tor inhibition suppresses Src-induced tumorigenesis.

3. Methionine is important for Tor signaling and Src-induced tumorigenesis.

Based on 1 and 2, we feel it was reasonable to describe that Src-p38-Tor signaling regulates Src-induced tumorigenesis. What was not clear was how Src-p38-Tor signaling crosstalks with methionine-Tor signaling and also how Src-p38 regulates Tor. The link between methionine and Tor was not strong either. As the reviewers see below, we spent considerable time and effort to address these points in this revision. What we eventually elucidated is a rather complicated mechanism, but at least it provides a clearer picture of what’s occurring during Src oncogenesis. The following is what we found.

Regarding methionine-Tor signaling, we found that Sam synthetase, which makes a methyl donor SAM from methionine, is important for Tor activation and Src-induced tumorigenesis (Figure 7 I-K). We also found that Samtor, a Sam sensor that regulates Tor, is important for Src-induced Tor regulation (Figure 7—figure supplement 2 D-F). These findings strengthen the link between methionine and Tor during Src-mediated oncogenesis.

Regarding how Src regulates methionine-Tor signaling, we found that Src promotes uptake of methionine in the tumor and also conversion of methionine to SAM (methionine flux) (Figure 8A-C). However, this was independent of p38 (Figure 8D-F). This indicates that Src could potentially activate methionine-Tor signaling through activation of methionine metabolism (uptake and flux) in a p38-independent manner.

So, a question is how Src-p38 contributes to Tor activation, since clearly Src-p38 signaling activates Tor (Figure 5A-B). Although previously it was shown that p38 activates Tor, its mechanism was not clear ^1^. To address this point sincerely, we started a new collaboration with Dr Diana Ho at Harvard, who previously performed RNAseq of the *vg>Src42A CA* wing disc ^2^. We surveyed expression levels of potential Tor regulators and selected genes that are affected by *Src* expression, including amino acid transporters and GATOR complexes. GATOR complexes regulate Tor through Rag GTPases ^3,4^. We examined whether their expression is regulated by Src in a p38-dpendent manner using RT-qPCR. We found that among the amino acid transporters and GATOR complex components examined, only *pathetic* (*path*), a SLC36 amino acid transporter that can transport multiple amino acids, was significantly induced by Src in a p38-dependent manner (Figure 8—figure supplement 1D-E). Since Path can mediate amino acids-mediated Tor activation ^5,6^, we speculate that Src-p38 could regulate Tor potentially through Path-mediated uptake of non-methionine amino acids.

We acknowledge that our data on Path’s role during Src tumorigenesis are indirect and at best suggestive, so we included the data of Path in discussion and carefully discussed this possible mechanism in the manuscript.

Our findings on the role for Src-p38 signaling in Tor activation is consistent with the previous findings that p38 is necessary for Tor activation in response to amino acids ^1^. Our suggestion of the role for non-methionine amino acids in Tor regulation, at a glance, may seem contradictory from our amino acid subtraction data, which showed that only subtraction of methionine could affect survival over the Src tumor (Figure 7B). But we speculate that because of Path’s function to uptake multiple amino acids, which provides redundancy, single amino acid subtraction experiments elucidated only the function of methionine. In a future study, it will be interesting to investigate effects of subtracting multiple amino acids at the same time.

Here we’d like to emphasize and clarify that the original purpose of this paper was to find the mechanism by which Src induces both cell death and proliferation. But somehow, while we explore the mechanism of Src-mediated Tor regulation and proliferation during revision, this part became unexpectedly and unbalancingly huge. But, we still feel that our original, simple schematic that explains the mechanism of cell death/proliferation is valid (Figure 8G). In this revised manuscript, we also included a schematic showing a detailed possible mechanism by which Src regulates methionine-Tor signaling (Figure 8—figure supplement 1F).

2. The link between the genetic experiments and nutrition, for example the role of methionine, is quite interesting, but is not so well developed and the reasoning behind the experiments is difficult to follow. It seems as though the logic is that diluting amino acids decreases the tumor growth, and that adding the amino acids back suppresses the rescue. Addition of amino acid, without methionine, limits tumor growth to a degree and increases survival compared to EAAs, though the magnitude of the rescue is difficult to interpret (e.g. 7.5 % vs 15 % survival, with quite large variance). Adding methionine alone clearly limits survival of flies with tumor, though it is not clear that block is caused by TOR alone. These latter metabolic experiments are fairly difficult to interpret, and the link between Src-Slpr-p38 and TOR and methionine is not as clear as the work presented earlier in the paper.

Regarding the link between the genetic experiments and nutrition manipulation, please see the response above.

Regarding the variance of the survival rate, we feel it’s an intrinsic property of physiological experiments like ours. Because of the variance in these physiological experiments, we adopted valid statistical analyses. All information of the statistical analyses were included in table 3.

3. Are there regional differences within the wing disc (Figure 1) where proliferation (pouch) vs apoptosis (edges) observed more commonly? How would such a difference fit into the model? Clonal analysis could shed light on this issue of autonomy

Let us clarify where apoptosis and proliferation occur. We quantified both autonomous and non-autonomous proliferation/death in this revised manuscript (Figure 1B, 1D and Figure 4—figure supplement 1D, 1G).

Regarding cell death, it’s almost exclusively cell autonomous, with a slight non-cell autonomous effect, which is consistent with the similar JNK activation pattern (Figure 2C). On the other hand, Src provokes cell proliferation both autonomously and non- autonomously, which is consistent with the similar p38 activation pattern (Figure 4A). We presume why pH3 looked more dominant in the pouch for the reviewer is likely because there are high, basal background signals of pH3 outside of the *vg-gal4* region while cell death is relatively restrictive to the *vg-gal4* region without basal background signals.

One thing we noticed during experimentation is that when we induced massive overgrowth by combining Src activation and JNK inhibition, the massive growth always occurs at the “side” region. We speculate that this “side region” may correspond to the previously identified “hot spots” ^7^

Regarding the clonal analysis, Src clone cells induce strong cell competition ^8^ and clonal analyses are intrinsically unreproducible and unpredictable, thus in this study we mainly stick to *vg-gal4*. This approach with *vg-gal4* and *UAS-Src42A CA* has been previously adopted by the group of Spyros Artavanis-Tsakonas ^2^.

Reviewer #2:In this paper, Nishida et al. address a long-standing and intensely-investigated question, namely how are cell proliferation and cell death controlled to modulate tissue growth in animals. They specifically investigate the effects of Src overexpression in imaginal tissues (which leads to both increases cell proliferation and cell death) and they make two key observations: (a) they identify the kinase Slpr as an effector of Src in the control of both cell proliferation (via p38) and cell death (via JNK), (b) they show that the effects of p38 are mediated via p38 and that dietary methionine can control TOR and modulate Src-mediated cell proliferation.Overall, I liked this paper. I think the two main findings will be of interest to, not just *Drosophila* researchers, but also cell, developmental and cancer biologists interested in the regulation of tissue growth.I have two main comments/questions:1. Are the effects on cell proliferation and cell death strictly autonomous? It's hard to tell clearly from Figure 1, especially since the choice of driver of vg-Gal4 doesn't have clearly demarcated expression boundaries (compared to, say, en-GAL4 or ap-GAL4). This is import since, if the proliferation is a result of cell-death induced compensatory proliferation, then the interpretations of the data in the paper would be very different than if everything was strictly autonomous.

This has an overlap with the reviewer 1’s comment (Major point 3).

In brief, Src induces cell death and proliferation both cell autonomously and non-cell autonomously. But, cell death is more cell autonomous while cell proliferation is rather both autonomous and non-autonomous. In this paper, we almost exclusively focused on the cell autonomous phenotype and manipulated genes autonomously.

The reviewer is speculating that non-autonomous cell death may lead to more autonomous cell proliferation due to compensatory proliferation. Theoretically that is possible, but we think it is unlikely because of the following reason. We quantified the effect of *slpr* RNAi to non-cell autonomous JNK activation and cell death. s*lpr* knockdown does not affect JNK or cell death non autonomously (Figure 4—figure supplement 1E-G) but affects cell autonomous proliferation (Figure 3A-D). this strongly suggests cell autonomous proliferation is due to the autonomous signaling events, not due to non-cell autonomous cell death.

We would also like to note that the boundary of *vg-gal4* is relatively clear in a normal situation, but *Src* expression makes the expression boundary less clear likely due to cell death. According to literature, this Src-induced hazy boundary seems to occur even with clonal analyses ^8^ or another gal4 driver such as *ptc-gal4*^10^.

2. The link between p38 and TOR is interesting, but it relies on using measurement of phospho-4EBP as a readout of TOR signaling. These data are certainly very clean, however, it’s important to note that total levels of 4EPB are often induced upon stress signaling. Hence, it's important to be sure that the increase in phospho-4EBP signal does reflect an increase in the amounts of phosphorylated 4EBP (a TOR effect) vs simply an increase in total 4EBP (a TOR-independent, stress effect). A recent paper from the Karpac lab (Cell Reports, 2020) had some nice immunostaining with a non-phospho 4EBP, which would be a useful control in this manuscript. Alternatively, the induction of TOR could be confirmed with a phospho-S6k or phospho-S6 antibody (western on discs or immunostaining). The genetic expts with TOR certainly support a role for TOR in modulating Src function, but they could equally be interpreted as TOR signaling being a parallel pathway or TOR being a downstream Src pathway.

We thank this reviewer for this useful comment. By striking a new collaboration with Dr Jongkyeong Chung, we included data of pS6. Src induces phosphorylation of S6 in addition to p4EBP(Figure 5—figure supplement 1B-C).

Reviewer #3:The authors examine how an oncogene, Src, can activate both proliferation and apoptosis during oncogenesis. The answer is surprisingly simple. In the case of Src, it activates both pro-apoptotic (JNK) and proliferative (p38) MAPK pathways and upstream of this, the authors identify the MPKKK slpr as the lynchpin in this pathway. Src also activates Erk-type MAPK, but according to the authors, this has no effect for tumorigenesis. The authors then focus on the role of p38 and found that p38 controls Tor signaling. Interestingly, the nutrition status does influence Src-induced Tor signaling and tissue growth. To further delineate this observation, the authors tested each amino acid and found specifically a requirement of Methionine for Src-induced tumor growth. They also observed that the Methionine concentration in the hemolymph of these Src-bearing larvae is lower compared to controls, presumably because the Src-tumor tissue absorbs more of it. Overall, the authors present a model that combines several signaling pathways and metabolism for Src-induced tumor growth.This is an interesting manuscript, but I do have some comments and suggestions for improvement.1. What is the nature of Src42A CA? How does it promote dominant activity to Src?

We failed to provide sufficient explanation. We apologize for this. We specified Src 42A CA is “*Src42A* with an amino acid substitution of Tyr^511^ to Phe , which is refractory to inactivating phosphorylation by Csk ^11^” in text.

2. Provide a complete picture of the Src CA suppressor screen in which they found slpr. Were there other suppressors and if so, why did they focus on slpr?

Our original screen focused on factors related to cell death, trying to find a mediator of Src-induced cell death. We included a list of RNAis we used (Figure 2—figure supplement 1A-B, table 1). Among RNAis we checked, only *slpr* RNAi demonstrated both high survival and recovered wings. Some of other RNAis enhanced survival but not the wing phenotype (Figure 2—figure supplement 1A-B), suggesting these should affect non-tissue autonomous event.

3. Present the results from the screen of the other MPKKK. Did they also screen the MPKK and other upstream components in the MPK pathway (Msn, Traf1, Traf2, Rho1)? This might tell us how Src activates Slpr.

At least, based on the survival assay, other JNKKK (dTAK1, Mekk1, Ask1, Wnd) were not hits. Msn, traf1, Traf2, msn or Rho1 were not positive hits either. But again, we’d like to emphasize a caveat of negative data in RNAi screening due to low efficiency of knockdown.

4. How does Src activate Slpr? Is any of the domains of Slpr (SH2, SH3, kinase) required for this activation?

This is related to the point 3 above. Among the potential mediators of Src-Slpr signaling, we found that inhibition of Shark, which is a non-receptor tyrosine kinase of the syk family, could suppress the wing phenotype by Src (Author response image 1). Shark could potentially function downstream of Src ^12^ To solidify and expand a story on Shark, we feel it’ll require much more experimentation. Importance of each domain of Slpr has been extensively investigated previously ^13,14^, and we did not focus on these domains in this study. We acknowledge that to clarify what mediates Src-Slpr signaling and which domain of Slpr is involved in the process could be an important future question.

5. How many different slpr RNAi lines were tested? Are the results verified with several independent lines? Do slpr mutants suppress Src42ACA?

We used two RNAis (Figure 2 and Figure 3—figure supplement 1A-D). We also checked knockdown efficiency of both RNAis by RT-qPCR (Figure 2—figure supplement 1C and Figure 3—figure supplement 1D). Slpr mutant is homozygous lethal, thus we decided not to use it in this manuscript.

6. Verify the Erk-MAPK (rl) RNAi line(s).

The *erk* RNAis we used had been previously validated (Singh, Liu, Zhao, Zeng, & Hou, 2016). Erk inhibition does not suppress proliferation (Fig 4-figure supplement 1I), which is in a clear contrast to p38 inhibition. We also verified an efficiency of an *erk* RNAi using the rough eye phenotype induced by Ras activation (Fig4-figure supplement 1J). To the best of our knowledge, there has been no Erk DN generated in *Drosophila*.

7. The data presentation needs improvement. The discs are too small and often the relevant staining (PH3, cDcp1, TRE-RFP, etc.) is hard to judge. This is also particularly important for the non-cell autonomy of Erk and p38 activity in Figure 4A-B. Some convincing close-up pictures could help.

We included magnified pictures in the revised manuscript. We also included quantification of almost all data.

8. There is no comprehensive description of how the quantitation of the staining was conducted. Was the quantitation done in the region of overgrowth? How was the volume of GFP+ tissue determined? Were equivalently sized regions used and how was their level of fluorescence assessed? Was there some internal control to normalize for differences in general staining between regions of interest and region where no change would be expected (ie a part of the disc outside of the vg-Gal4 area)?

We included this information in the method section.

9. Figure 1A: quantify the cDcp1 staining.

We quantified the data as requested (Figure 1D).

10. Figure 2C: slpr RNAi seems to suppress TRE-RFP, but not overgrowth. In fact, the slpr RNAi disc (lower panels) is significantly larger than the src42ACA disc (middle panels) including the GFP+ area. That is inconsistent with the authors model. Explain. Please quantify GFP+ areas and overall disc size.

As the reviewer rightfully noticed, *slpr* RNAi suppresses TRE-RFP, that is, cell death. Because of the suppression of cell death, *slpr* RNAi makes discs bigger than the Src42A CA disc, where death continuously occurs. Compared to the real overgrowth phenotype by Src activation and JNK inhibition, the effect of *slpr* RNAi to Src disc is much milder because it suppresses both cell death and proliferation. We quantified all data as requested.

11. Figure 3B: Perform PH3 labeling in vg>Src42ACA JNKDN slprRNAi background. I would like to know if the overgrowth and PH3 labeling in Src42ACA JNKDN is dependent on Slpr. The authors imply this, but don't show it.

We performed this experiment as requested (Figure 3—figure supplement 1G-I).

12. Figure 5A-B: is the 4EBP phosphorylation in vg>Src42CA JNK DN discs depended on Slpr, i.e. does slpr RNAi block it?

We now included data showing that *slpr* RNAi blocks S6 phosphorylation in *vg>src42A CA, JNK DN* discs (Figure 5—figure supplement 1B-C).

13. In the Discussion, the authors mention that "methionine regulates Tor activation". I don't think this direct link was established in this paper.

We mainly focused on this point in this revision. Please see the response to the reviewer 1.

14. Is the a SAMTOR homolog in flies? If so, test it in the Src42ACA screen.

Yes. The paper by the Sabatini lab demonstrated that SAMTOR works in the fly cells ^15^. In this revised manuscript, we inhibited samtor. As expected, it enhanced tor (Figure 7—figure supplement 2D-F), but puzzlingly it suppressed proliferation. We speculate that this growth suppression is likely attributed to Samtor’s predicted methyltransferase function, which may not be related to Tor signaling.

References:

1 Cully, M. *et al.* A role for p38 stress-activated protein kinase in regulation of cell growth via TORC1. *Mol Cell Biol* 30, 481-495, doi:10.1128/MCB.00688-09 (2010).

2 Ho, D. M., Pallavi, S. K. and Artavanis-Tsakonas, S. The Notch-mediated hyperplasia circuitry in *Drosophila* reveals a Src-JNK signaling axis. *eLife* 4, e05996, doi:10.7554/*eLife*.05996 (2015).

3 Kim, J. and Guan, K. L. mTOR as a central hub of nutrient signalling and cell growth. *Nat Cell Biol* 21, 63-71, doi:10.1038/s41556-018-0205-1 (2019).

4 Sabatini, D. M. Twenty-five years of mTOR: Uncovering the link from nutrients to growth. *Proc Natl Acad Sci U S A* 114, 11818-11825, doi:10.1073/pnas.1716173114 (2017).

5 Newton, H. *et al.* Systemic muscle wasting and coordinated tumour response drive tumourigenesis. *Nat Commun* 11, 4653, doi:10.1038/s41467-020-18502-9 (2020).

6 Goberdhan, D. C., Meredith, D., Boyd, C. A. and Wilson, C. PAT-related amino acid transporters regulate growth via a novel mechanism that does not require bulk transport of amino acids. *Development* 132, 2365-2375, doi:10.1242/dev.01821 (2005).

7 Tamori, Y., Suzuki, E. and Deng, W. M. Epithelial Tumors Originate in Tumor Hotspots, a Tissue-Intrinsic Microenvironment. *PLoS Biol* 14, e1002537, doi:10.1371/journal.pbio.1002537 (2016).

8 Enomoto, M. and Igaki, T. Src controls tumorigenesis via JNK-dependent regulation of the Hippo pathway in *Drosophila*. *EMBO Rep* 14, 65-72, doi:10.1038/embor.2012.185 (2013).

9 Singh, S. R., Liu, Y., Zhao, J., Zeng, X. and Hou, S. X. The novel tumour suppressor Madm regulates stem cell competition in the *Drosophila* testis. *Nat Commun* 7, 10473, doi:10.1038/ncomms10473 (2016).

10 Vidal, M., Larson, D. E. and Cagan, R. L. Csk-deficient boundary cells are eliminated from normal *Drosophila* epithelia by exclusion, migration, and apoptosis. *Dev Cell* 10, 33-44, doi:10.1016/j.devcel.2005.11.007 (2006).

11 Tateno, M., Nishida, Y. and Adachi-Yamada, T. Regulation of JNK by Src during *Drosophila* development. *Science* 287, 324-327, doi:10.1126/science.287.5451.324 (2000).

12 Biswas, R., Stein, D. and Stanley, E. R. *Drosophila* Dok is required for embryonic dorsal closure. *Development* 133, 217-227, doi:10.1242/dev.02198 (2006).

13 Garlena, R. A., Gonda, R. L., Green, A. B., Pileggi, R. M. and Stronach, B. Regulation of mixed-lineage kinase activation in JNK-dependent morphogenesis. *J Cell Sci* 123, 3177-3188, doi:10.1242/jcs.063313 (2010).

14 Neisch, A. L., Speck, O., Stronach, B. and Fehon, R. G. Rho1 regulates apoptosis via activation of the JNK signaling pathway at the plasma membrane. *J Cell Biol* 189, 311-323, doi:10.1083/jcb.200912010 (2010).

15 Gu, X. *et al.* SAMTOR is an S-adenosylmethionine sensor for the mTORC1 pathway. *Science* 358, 813-818, doi:10.1126/science.aao3265 (2017).